# Fully exposed Pt clusters for efficient catalysis of multi-step hydrogenation reactions

Yang Si[1,2,10], Yueyue Jiao[3,4,5,10], Maolin Wang [6], Shengling Xiang[7], Jiangyong Diao[2] ✉, Xiaowen Chen[1,2], Jiawei Chen[1,2], Yue Wang[2,8], Dequan Xiao [9], Xiaodong Wen [3,4], Ning Wang [7], Ding Ma [6] ✉ & Hongyang Liu [1,2] ✉

For di-nitroaromatics hydrogenation, it is a challenge to achieve the multi-step hydrogenation with high activity and selectivity due to the complexity of the process involving two nitro groups. Consequently, many precious metal catalysts suffer from low activity for this multi-step hydrogenation reaction. Herein, we employ a fully exposed Pt clusters catalyst consisting of an average of four Pt atoms on nanodiamond@graphene ($Pt_n$/ND@G), demonstrating excellent catalytic performance for the multi-step hydrogenation of 2,4-dinitrotoluene. The TOF ($40647\ h^{-1}$) of $Pt_n$/ND@G is significantly superior to that of single Pt atoms catalyst, Pt nanoparticles catalyst, and even all the known catalysts. Density functional theory calculations and absorption experiments reveal that the synergetic interaction between the multiple active sites of $Pt_n$/ND@G facilitate the co-adsorption/activation of reactants and $H_2$, as well as the desorption of intermediates/products, which is the key for the higher catalytic activity than single Pt atoms catalyst and Pt nanoparticles catalyst.

Catalytic hydrogenation of nitroarenes is an important industrial reaction for the production of aromatic amines, which play a fundamental role in the manufacture of synthetic intermediates of various chemicals[1–3]. Amongst these, one of the commonly used processes is the hydrogenation of 2,4-dinitrotoluene (2,4-DNT) to 2,4-diaminotoluene (2,4-DAT), with more than 1 million tons produced globally per year[4]. The desired 2,4-DAT product is a value-added intermediate, which is predominantly used for the production of polyurethane foams, adhesives, and elastomers[5,6]. For the 2,4-DNT hydrogenation,

the presence of two nitro groups demands a multi-step and multi-electron hydrogenation pathway that can overcome various side reactions. Meanwhile, the conversion of the first nitro group to the amine group hinders further hydrogenation of the second nitro group[7–9]. As a result, chemoselective hydrogenation of 2,4-DNT is still a challenge, as various side products (e.g., nitroso compounds, hydroxylamine, and azoxy and azo compounds) could be observed[7,10]. Supported platinum group metal catalysts have been widely applied for the hydrogenation of nitroarenes due to their good catalytic

[1]School of Materials Science and Engineering, University of Science and Technology of China, Shenyang 110016, PR China. [2]Shenyang National Laboratory for Materials Science, Institute of Metal Research, Chinese Academy of Sciences, Shenyang 110016, PR China. [3]State Key Laboratory of Coal Conversion, Institute Coal Chemistry, Chinese Academy of Sciences, Taiyuan 030001, PR China. [4]National Energy Center for Coal to Clean Fuel, Synfuels China Co., Ltd, Beijing 100871, PR China. [5]The University of Chinese Academy of Sciences, Beijing 100049, PR China. [6]Beijing National Laboratory for Molecular Sciences, New Cornerstone Science Laboratory, College of Chemistry and Molecular Engineering, Peking University, Beijing 100871, PR China. [7]Department of Physics and Center for Quantum Materials, Hong Kong University of Science and Technology, Kowloon, Hong Kong SAR, PR China. [8]Department of Chemistry, Liaoning University, 66 Chongshan Road, Shenyang, Liaoning 110036, PR China. [9]Center for Integrative Materials Discovery, Department of Chemistry and Chemical Engineering, University of New Haven, West Haven, CT 06516, USA. [10]These authors contributed equally: Yang Si, Yueyue Jiao. ✉e-mail: jydiao@imr.ac.cn; dma@pku.edu.cn; liuhy@imr.ac.cn

activity[11,12]. But, they usually suffered from not only the high cost and relative scarcity but also low activity and low atomic efficiency for di-nitroaromatics hydrogenation. Hence, it is urgent to develop efficient catalysts with high activity and selectivity, high metal utilization, and low cost for this multi-step hydrogenation reaction.

In recent years, single-atom catalysts (SACs), as a novel frontier in heterogeneous catalysis, have attracted extensive attention from both academia and industry[13–15]. With atomic metal dispersion, SACs maximize metal utilization with unique electronic properties, which are particularly important for the efficient application of precious metal resources[16]. Nevertheless, the dissociation of $H_2$ in the single active site of SACs must proceed via a heterolytic pathway with a significantly higher barrier than that of typical homolytic dissociation in nanosized metal surfaces, especially when the single-atom sites are pre-covered by other reactants[17]. Therefore, it is hard to optimally adjust the adsorption of reactants and intermediates on the solo sites of SACs simultaneously due to the lack of neighboring metal atoms, resulting in its limits for certain reactions, especially multi-step reactions[18–21]. Generally, optimizing the adsorption of $H_2$, reactants, and intermediates for multi-step reactions is the key to the catalyst design. As a cross-dimensional extension to the concept of SACs, fully exposed cluster catalysts (FECCs) emerge as a type of catalyst in which all metal atoms on the catalyst are fully exposed on the surface of the active sites without bulk phase structure. This design guarantees full atomic utilization during the reaction, providing catalytic sites with multiple metal atoms[22,23]. FECC is so highly dispersed, allowing all the metal atoms to be available for the adsorption and transformation of reactants. Supported FECCs have two distinct advantages: their ultrasmall size (typically below 1 nm) eliminates undesired bulk atoms and reduces the average coordination number of metal atoms, while a small contact angle between the metal and support enhances interaction and ultimately increases cluster stability[22,24]. From this viewpoint, FECCs with multiple active sites have distinct advantages in overcoming the limitations of SACs for a multi-step reaction.

Herein, we fabricated fully exposed Pt clusters catalysts ($Pt_n$/ND@G) on nanodiamond@graphene (ND@G) for the multi-step hydrogenation of 2,4-DNT. Through a combination of aberration-corrected high-angle annular dark-field scanning transmission electron microscopy (AC-HAADF-STEM) and X-ray absorption fine structure (XAFS), the $Pt_n$/ND@G with an average of 4 atoms via Pt−C bonds has been identified. $Pt_n$/ND@G exhibited excellent catalytic activity for 2,4-DNT hydrogenation under room temperature: high TOF (40,647 $h^{-1}$), high yield (>99%), and good stability (5 cycles), surpassing Pt single atoms catalysts ($Pt_1$/ND@G), Pt nanoparticles catalysts ($Pt_p$/ND@G) supported on ND@G, or all other known catalysts[4,25]. Moreover, $Pt_n$/ND@G demonstrated outstanding catalytic performance in the multi-step hydrogenation of di-nitroarenes with various substitutional groups. DFT calculations and absorption experiments results suggested that the fully exposed Pt clusters with multiple active sites are beneficial to provide enough sites for the subsequent dissociation of $H_2$, and possess moderate adsorption ability towards intermediates and product, resulting in the exceptional catalytic performance for this multi-step hydrogenation reaction. The present study paves a new avenue for designing and developing novel catalysts for multi-step hydrogenation.

## Results

### Preparation and characterization of atomically dispersed Pt catalysts

The synthetic method of ND@G was based on our previous work, and the detailed preparation processes can be found in the "Methods" section [26,27]. Briefly, the ND@G was obtained by high-temperature calcination in an inert atmosphere for the pristine nanodiamonds. As shown by the high-resolution transmission electron microscopy (HRTEM) images (Fig. 1a), the ND@G material features a core-shell

structure with a nanodiamond core and defect-rich graphene shell. The abundant carbon defects of ND@G play a vital role in anchoring metal atoms, enabling ND@G to be more conducive to dispersing and stabilizing metal atoms than some oxides or carbon materials[24,28,29]. By modulating the Pt loading amount, we prepared two different atomically dispersed catalysts (0.5 wt% $Pt_n$/ND@G and 0.1 wt% $Pt_1$/ND@G). We then used AC-HAADF-STEM to analyze the structure and morphology of $Pt_n$/ND@G. As shown in Fig. 1b, many small Pt clusters are uniformly distributed on ND@G support with dominant species in the 0.2–0.5 nm range, and no remarkable formation of aggregates was observed. In Fig. 1c, the fully exposed Pt clusters (labeled in blue circles) without crystal structures were found, demonstrating that the Pt species were atomically homogeneously dispersed on the ND@G support. Furthermore, the Pt clusters are single-atomic-layer thick as shown in the high magnification STEM image and the corresponding intensity profile (see Fig. 1d, e), implying the fully exposed characteristics. The X-ray diffraction (XRD) pattern of the $Pt_n$/ND@G showed a series of peaks at 43.9° and 75.3°, which can be attributed to the characteristic peaks of diamond (PDF, #06-0675)[30]. The $Pt_n$/ND@G samples did not display diffraction peaks corresponding to Pt crystal, indicating the small size of Pt species. The $H_2$/$O_2$ titration measurement (Supplementary Table 1) showed the Pt dispersion of $Pt_n$/ND@G was 93%, indicating the almost fully exposed Pt clusters on the defect-rich graphene, consistent with the STEM results. For $Pt_1$/ ND@G (Supplementary Fig. 1), isolated Pt atoms were identified in a uniform dispersion on the ND@G surface. Meanwhile, no obvious diffraction peaks of Pt crystal were observed in the XRD pattern of $Pt_1$/ND@G.

To probe the electronic and coordination environment of the fully exposed Pt clusters, we performed synchrotron XAS on the Pt $L_3$-edge within both the extended X-ray absorption fine structure (EXAFS) regions and the X-ray absorption near-edge structure (XANES), and compared them to those of Pt foil and $PtO_2$. The Fourier transform (FT) of EXAFS (Fig. 2a–c) in the R space and the wavelet transformation (WT) of EXAFS oscillations (Fig. 2d–f) were performed to study the distinct coordination structure of Pt species. The $Pt_n$/ND@G sample exhibited a sharp peak at 1.6 Å and a relatively weak peak at 2.6 Å, which are assigned to the first coordination shell of Pt C/O and Pt-Pt, respectively. The trend can also be clearly resolved from the WT of Pt $L_3$-edge EXAFS oscillation results. The EXAFS data-fitting results (Supplementary Table 2) manifested that the Pt-Pt coordination number (CN) in $Pt_n$/ND@G was ~2.5, implying that the fully exposed Pt clusters have an average atomicity of four Pt atoms. For $Pt_1$/ND@G, only one peak centering about 1.8 Å was identified as the Pt−C/O bond, and no Pt−Pt scattering was detected, implying that the single Pt atom coordinates only with carbon and/or oxygen atoms. Moreover, the fine structure of Pt species (in single atoms and clusters) was studied by CO adsorption using in situ diffuse reflectance infrared Fourier transform spectroscopy (DRIFTS). The fresh $Pt_1$/ND@G samples displayed only a weak peak at ~2100 $cm^{-1}$, assigned to the vibration of CO absorbed on isolated $Pt_1$ atoms[31,32]. In contrast, the main absorption band of CO (~2046 $cm^{-1}$) exhibited a significant red shift for the CO linearly absorbed on Pt sites of the fresh $Pt_n$/ND@G samples (Supplementary Fig. 2), indicating the presence of Pt clusters[33].

From the XANES results, the valence states of $Pt_n$/ND@G were revealed. As displayed in Fig. 2g, the Pt $L_3$-edge XANES curves indicated that the adsorption edge energy ($E_0$) in $Pt_n$/ND@G is located between Pt foil and $PtO_2$. This suggests that the Pt clusters carry a positive charge owing to the electron transfer from Pt to the matrix[34,35], demonstrating the interaction between Pt species and the ND@G support. As expected, the absorption edge energy of $Pt_1$/ND@G was obviously close to that of $PtO_2$, suggesting that the Pt species in $Pt_1$/ ND@G was more positively charged than that in $Pt_n$/ND@G. To further examine the electronic states of the catalyst, the X-ray photoelectron spectroscopy (XPS) analysis was performed (Fig. 2h, i). There are two peaks in the high-resolution Pt 4$f$ XPS spectrum of $Pt_n$/ND@G with the

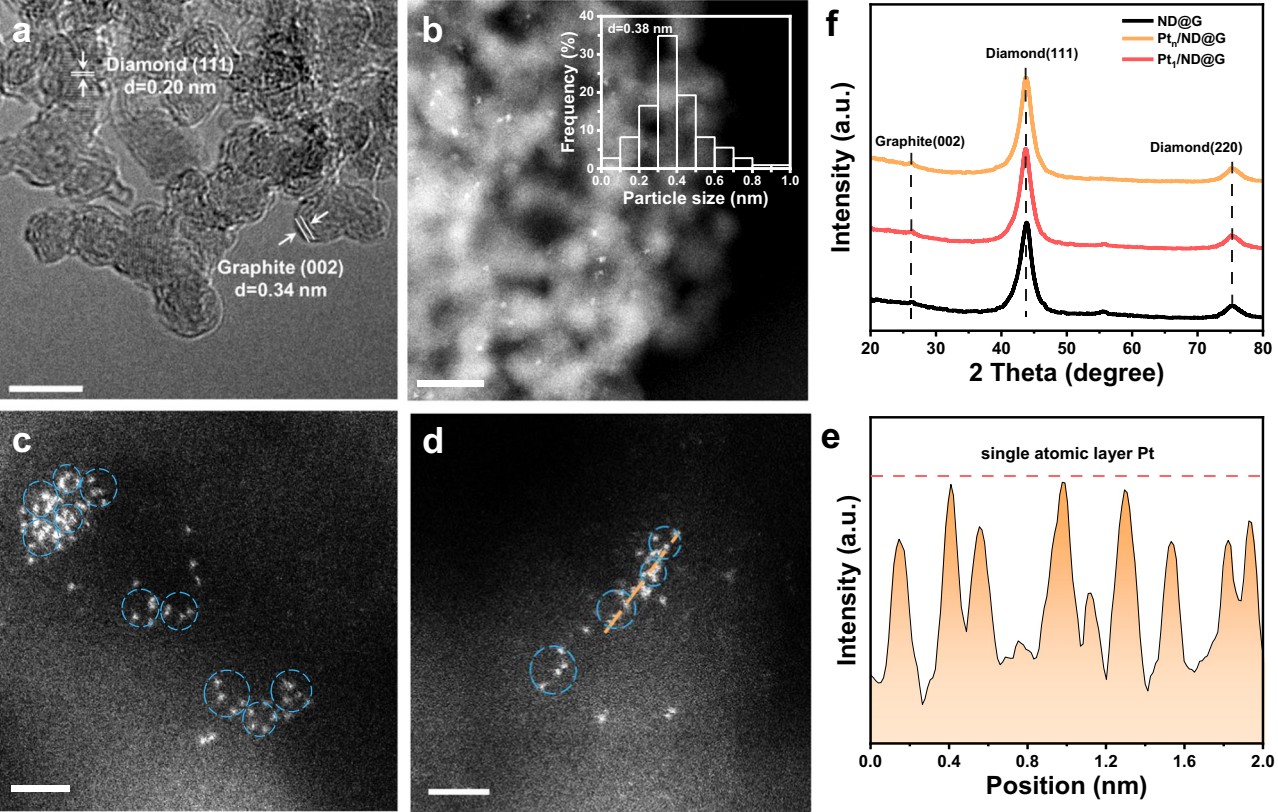

**Fig. 1 | Microscopic characterization of Pt$_n$/ND@G and Pt$_1$/ND@G. a** HRTEM image of the ND@G. Scale bar, 5 nm. **b** AC-HAADF-STEM images of Pt$_n$/ND@G at low magnification. Scale bar, 10 nm. **c**, **d** AC-HAADF-STEM images of Pt$_n$/ND@G at high magnification. Scale bar, 5 nm. **e** intensity profile along the line of Pt$_n$/ND@G. **f** XRD patterns of Pt$_n$/ND@G and Pt$_1$/ND@G. (Pt clusters in images are highlighted by blue circles.).

binding energies of 72.2 eV and 75.3 eV, associated with Pt 4$f_{7/2}$ and Pt 4$f_{5/2}$ peaks, respectively. The peaks of Pt$_1$/ND@G at 72.8 eV and 76.1 eV were ascribed to Pt 4$f_{7/2}$ and Pt 4$f_{5/2}$ peaks, showing a positive shift as compared with that of Pt$_n$/ND@G. The shift of the binding energy to high values suggests that the electronic states increased of Pt species from Pt clusters to Pt single atoms, consistent with the XANES result.

**Catalytic performance for the hydrogenation of 2,4-DNT**

We investigated the catalytic performances of Pt$_n$/ND@G for the hydrogenation of 2,4-DNT, together with a variety of other substituted nitroarenes. The possible reaction pathways of the hydrogenation of 2,4-DNT are proposed in Fig. 3a. First, we performed the solvent screening to systematically explore the solvent effect. Protic solvents, including methanol, ethanol, and isopropanol, polar aprotic solvents, including THF, ethyl acetate, and 1,4-dioxane, and nonpolar n-hexane, benzene, as well as toluene were applied for the hydrogenation of 2,4-DNT (Fig. 3b). The reaction in protic solvents (i.e., methanol, ethanol, isopropanol) achieved the complete conversion and high yields, and methanol was the best solvent in our catalytic system, probably as the alcohols with short-carbon-chain length could improve H$_2$ solubility[36]. In most cases, the conversions and yields were significantly lowered for the aprotic solvents (polar or nonpolar). These results suggested that the protic solvents seem to play a dominant role in the nitroarenes hydrogenation, similar to previous reports[37]. Temperatures and H$_2$ pressure were investigated, and Pt$_n$/ND@G demonstrated high catalytic performances even at room temperature (25 °C) and low pressure (1 MPa). Therefore, in the following, the catalytic hydrogenation of 2,4-DNT was examined under 1 MPa H$_2$ at 25 °C with methanol as solvent.

To study the advantages of ND@G support, a series of Pt catalyst samples (0.5 wt% Pt loading) were prepared and characterized by STEM (Supplementary Fig. 3) using various supports, including carbon

and oxide supports, and evaluated their catalytic activity under identical reaction conditions. As shown in Supplementary Fig. 3, clear Pt nanoparticles were visible in all catalysts through STEM images. Figure 3c illustrates the conversions of 2,4-DNT and the yields of 2,4-DAT for each catalyst after 2 h of reaction. Pt/G, Pt/ND, Pt/Al$_2$O$_3$, and Pt/TiO$_2$ exhibited high conversions (100%), whereas other catalysts demonstrated low conversions (Pt/AC and Pt/SiO$_2$, <25%). But all these catalysts had low yields (<60%). Pt$_n$/ND@G exhibited exceptional catalytic activity, significantly surpassing other Pt-based catalysts. The above results further indicate that the ND@G support is more efficient in dispersing and stabilizing metal atoms compared to oxides or carbon materials. Furthermore, the commercial 5 wt% Pt/C was used as a reference sample, which displayed well-dispersed Pt nanoparticles (Supplementary Fig. 4). As shown in Fig. 3c, the commercial Pt/C catalyst displayed lower activity compared to Pt$_n$/ND@G.

As shown in Fig. 3d, Pt$_n$/ND@G displayed an exceptionally high catalytic activity with a TOF of 40,647 h$^{-1}$, with a conversion of 100% and a yield of >99% towards 2,4-DAT after 2 h of reaction. In contrast, Pt$_1$/ND@G exhibited negligible activity, similar to the reported hydrogenation of nitroarenes using single-atom catalysts[19,38]. Meanwhile, the TOF of Pt$_n$/ND@G is about 60 times higher than that of Pt$_1$/ND@G. Additionally, the support of pure ND@G was inactive. Hence, the fully exposed Pt clusters, rather than Pt single atoms, are the active sites for the hydrogenation. The apparent activation energies (Supplementary Fig. 5) were calculated for Pt$_1$/ND@G and Pt$_n$/ND@G. The Arrhenius-type plots for Pt$_n$/ND@G showed a linear correlation, affording an apparent activation energy of 43.3 kJ/mol. Clearly, the apparent activation energy of Pt$_n$/ND@G is remarkably lower than that of Pt$_1$/ND@G (86.1 kJ/mol), consistent with the fact that Pt$_n$/ND@G was the most active catalysts. Compared to the known catalysts in literature, Pt$_n$/ND@G exhibits superior catalytic

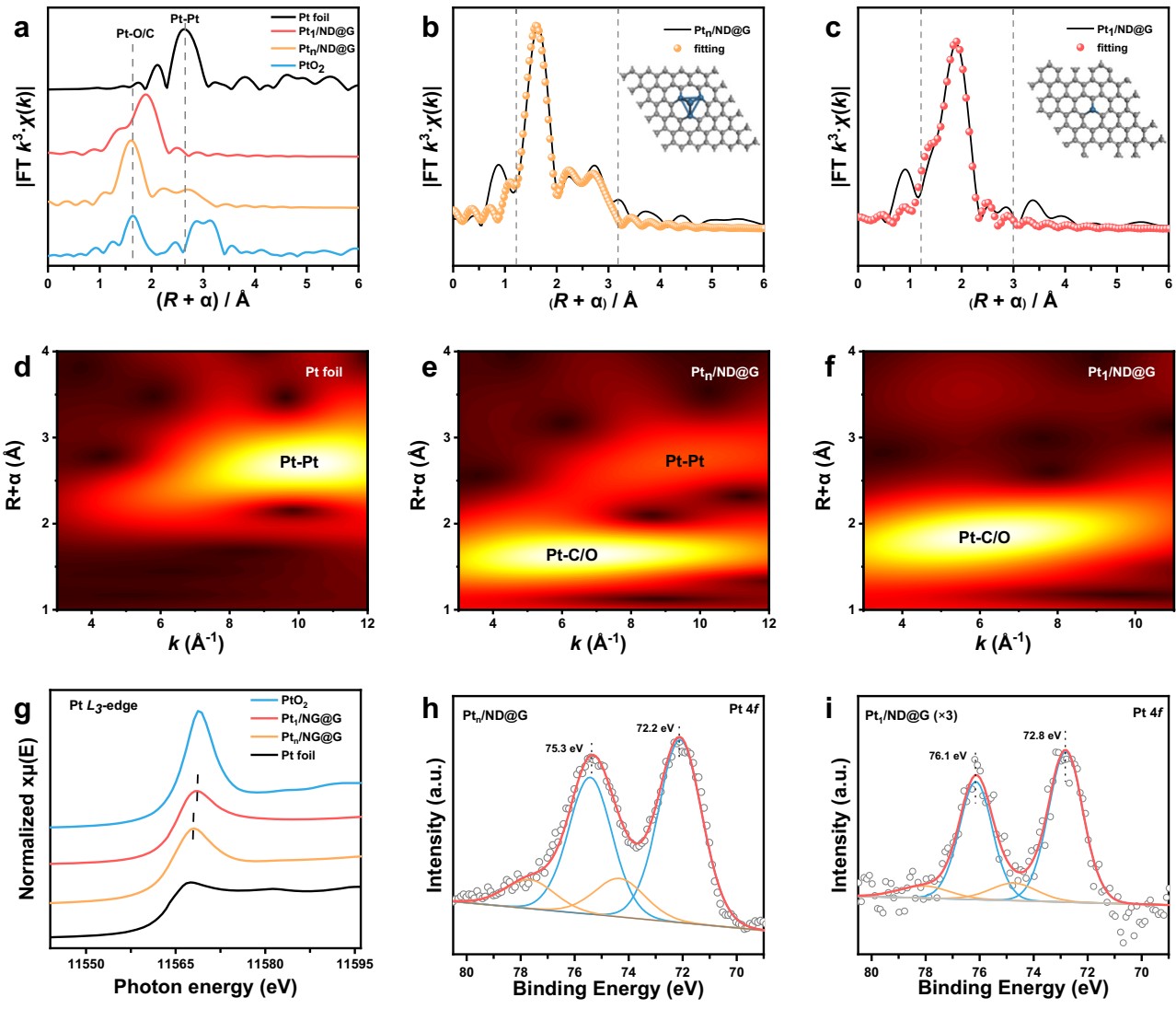

**Fig. 2 | Synchrotron XAFS and XPS measurements of catalysts. a** EXAFS Fourier transform (FT) $k^3$-weighted $\chi(k)$ function spectra. **b**, **c** EXAFS fitting results in R space for $Pt_n$/ND@G and $Pt_1$/ND@G. **d**–**f** WT analysis of $Pt_n$/ND@G, $Pt_1$/ND@G, and Pt foil. **g** Pt $L_3$-edge XANES spectra of $Pt_n$/ND@G and $Pt_1$/ND@G. **h**, **i** Pt $4f$ XPS spectra of $Pt_n$/ND@G and $Pt_1$/ND@G.

performance under mild reaction conditions. (Supplementary Table 3).

For comparison, the Pt nanoparticles catalysts ($Pt_p$/ND@G) with 1.0 wt% Pt loading were synthesized using the impregnation method for the hydrogenation of 2,4-DNT. As shown in Supplementary Fig. 6, the nanoparticles in $Pt_p$/ND@G showed distinct lattice fringes of Pt crystal and an average particle diameter distribution of 2.73 nm. The STEM image displayed a uniform lattice spacing of 0.226 nm, corresponding to the Pt (111) plane. As shown in Supplementary Fig. 7, the obvious diffraction peak in the XRD profile of $Pt_p$/ND@G was attributed to the Pt (111) surface of nanoparticles, which agrees with the STEM image. The Pt dispersion in $Pt_p$/ND@G is 65.8% as determined by the $H_2$/$O_2$ titration measurement (Supplementary Table 1). $Pt_p$/ND@G delivered a TOF of 26,750 $h^{-1}$ and a yield of only 56% and exhibited an apparent activation barrier of 47.5 kJ/mol (Fig. 3d and Supplementary Fig. 5). Thus, the catalytic activity of the $Pt_p$/ND@G is lower than that of $Pt_n$/ND@G.

The durability of catalysts was evaluated. The $Pt_n$/ND@G can be used for 5 cycles at low conversion without any loss in activity (Supplementary Fig. 8). The STEM images and ICP results of $Pt_n$/ND@G after the reaction indicated the absence of any aggregation and leaching, demonstrating its outstanding structural stability

during the reaction (Supplementary Fig. 9 and Supplementary Table 4).

We tested the hydrogenation for a set of di-nitroarenes with different substitutional groups. $Pt_n$/ND@G was used under room temperature, and the hydrogenation of these di-nitrobenzene derivatives delivered the desired products with excellent yields within several hours (Fig. 4, 1–6). Importantly, halogen-substituted di-nitroarenes underwent smooth conversion to haloaromatic amines without any dehalogenation (Fig. 4, 5–6). We explored a broad range of substituted nitroarenes. Other than nitrobenzene (Fig. 4, 7), $Pt_n$/ND@G also achieved outstanding yields for the hydrogenations of substituted nitrobenzenes containing electron-donor or electron-acceptor groups, as well as easily reducible groups like aldehyde and keto (Fig. 4, 8–16). Thus, $Pt_n$/ND@G was a versatile catalyst for the chemoselective hydrogenation of substituted nitroarenes.

## DFT calculations and mechanism insight

The hydrogenation of 2,4-DNT is a super-exothermic reaction, which is verified experimentally and theoretically[39,40]. And our DFT calculations also show the same result (Supplementary Table 5). This leads to the fact that once the reactants (2,4-DNT and $H_2$) can adsorb onto the active site, it will be a key factor for the reaction activity that whether

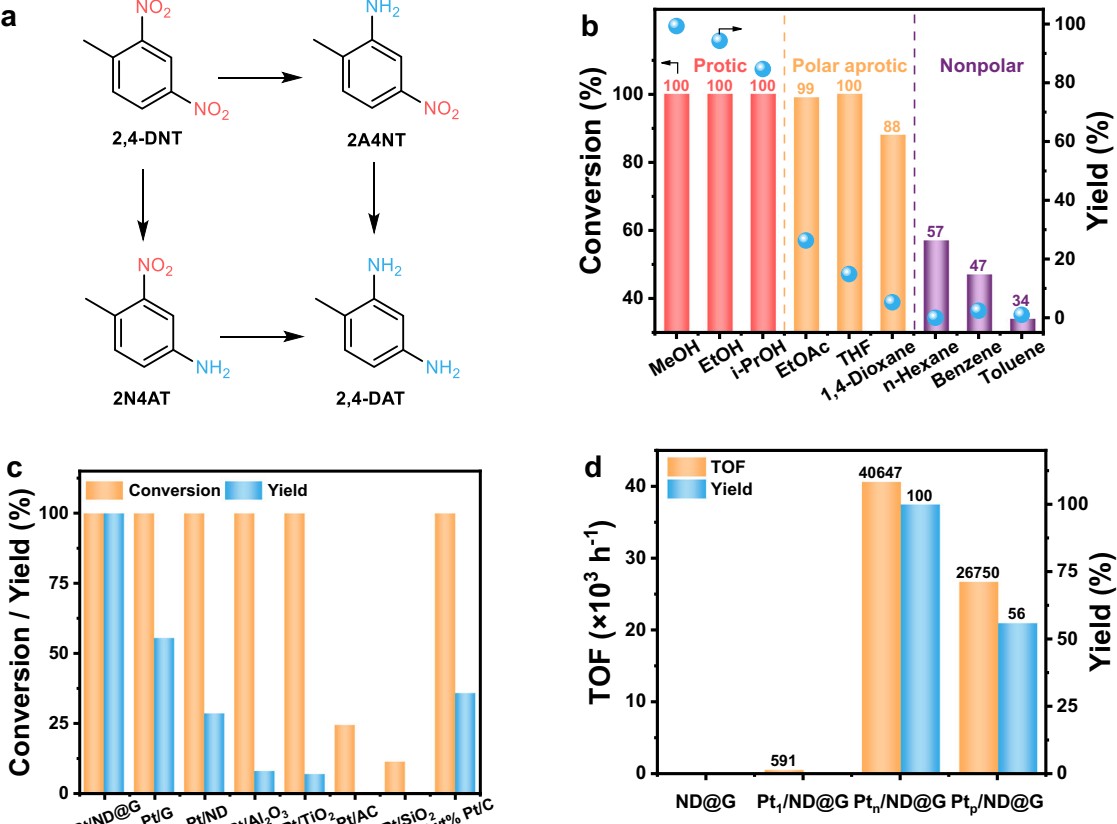

**Fig. 3 | Catalytic performance for the 2,4-DNT hydrogenation reaction. a** The possible reaction pathways for the hydrogenation of 2,4-dinitrotoluene, 2,4-DNT: 2,4-dinitrotoluene, 2A4NT:2-amino-4-nitrotoluene, 4A2NT:4-amino-2-nitrotoluene, 2,4-DAT:2,4-diaminotoluene. **b** Catalytic results on $Pt_n$/ND@G in various solvents (MeOH methanol, EtOH ethanol, i-PrOH isopropanol, EtOAc ethyl acetate, THF tetrahydrofuran). Reaction conditions: 5 mg of catalyst, 1 mmol of 2,4-DNT, 10 mL of solvent,1 MPa of $H_2$, 60 °C, and reaction time 20 min. **c** Catalytic

performance of Pt loaded on the series of supports (G graphene, ND nanodiamond, AC active carbon). Reaction conditions: the amount of Pt to the substrate is 0.0256 mol%, 1 mmol of 2,4-DNT, 10 mL of methanol,1 MPa of $H_2$, 25 °C, and reaction time 2 h. **d** TOF values of ND@G, $Pt_1$/ND@G, $Pt_n$/ND@G, and $Pt_p$/ND@G. Reaction conditions: the amount of Pt to the substrate is 0.0256 mol%, 1 mmol of 2,4-DNT, 10 mL of methanol,1 MPa of $H_2$, 25 °C, and reaction time 2 h.

the products and/or intermediates can desorb and release the active site successfully. To understand the difference in the hydrogenation among Pt single atoms, Pt clusters, and Pt metal particles. Based on the results of XAFS and STEM, we used $Pt_1$@Gr, $Pt_4$@Gr, and Pt(111) (Supplementary Fig. 10) as the models, respectively.

For the adsorption of 2,4-DNT, three groups (2-$NO_2$-, 4-$NO_2$-, and benzene-) can be attached to the catalyst surface, and the most stable adsorption configurations and the corresponding adsorption energies are shown in Fig. 5. On $Pt_4$@Gr, the adsorption energy by 2-$NO_2$- (−0.26 eV) is close to that by benzene- (−0.27 eV), and both are more negative than that by 4-$NO_2$- (−0.19 eV). On $Pt_1$@Gr, the adsorption by 2-$NO_2$- (−0.53 eV) is stronger than that by 4-$NO_2$- (−0.45 eV) or benzene- (−0.32 eV). On Pt(111) surface, the adsorption by 2-$NO_2$-, 4-$NO_2$- and benzene- has the adsorption energies of −0.18, −0.24, and −0.23 eV, respectively. $Pt_1$@Gr shows stronger adsorption to 2,4-DNT than $Pt_4$@Gr and on Pt(111) surface. The Bader analysis (Supplementary Fig. 10) reveals that the isolated Pt atom loses more electrons (−0.32 e) than the four Pt atoms (0.36 e, −0.18 e, −0.23 e, and −0.20 e) of $Pt_4$ clusters, which is consistent with the results of XAFS and XPS. The existence of a positively charged Pt single atom favors the preferentially adsorbing by −$NO_2$, consistent with the findings in the literature[41,42].

However, the catalytic activity of the hydrogenation of 2,4-DNT is not only related to the adsorption of 2,4-DNT, it may be also related to other reaction steps, including the adsorption and dissociation of $H_2$, the interaction between 2,4-DNT and $H_2$, and the occupancy and release of active sites. Further, we compared the adsorption and

dissociation of $H_2$ after the adsorption of 2,4-DNT on three catalyst models (Supplementary Table 6). On Pt(111) surface, the dissociation of $H_2$ is spontaneous, which is exothermic by 1.13 eV, implying a potentially high catalytic activity for 2,4-DNT. Compared with $Pt_4$@Gr, the adsorption of $H_2$ on $Pt_1$@Gr (0.01 vs. −0.16 eV) and its dissociation (0.18 vs. −0.87 eV) is more thermodynamically unfavorable. Therefore, it is difficult for a single Pt atom to connect with both $H_2$ and reactant during the reaction, and thus the hydrogenation activity of 2,4-DNT on $Pt_1$@Gr is significantly lower than that of $Pt_4$@Gr and Pt(111) surface. Further calculations were conducted to study the adsorption behavior of intermediates and products on $Pt_4$@Gr in comparison with Pt(111). The corresponding structures and adsorption energies were supplied in Supplementary Information (Supplementary Table 7–9). For 2A4NT and 2,4-DAT, both of them are in the configuration of benzene-groups on Pt4@Gr and on Pt (111). Compared Pt(111) with $Pt_4$@Gr, it is found that although $H_2$ dissociation is easier on Pt(111), the adsorption of 2,4-DAT is too strong (−1.57 vs. −0.74 eV, Supplementary Table 7), leading to accumulation of intermediates and products on active sites, consequently resulting in decreased activity.

To gain more insight into the disparity in hydrogenation activities between $Pt_n$/ND@G and $Pt_1$/ND@G, we employed in situ DRIFTS to monitor the changes of the infrared characteristic peaks of the nitro group on the catalyst samples before and after the introduction of hydrogen. Because 2,4-DNT is a solid at room temperature, it is difficult to diffuse 2,4-DNT to the catalyst surface as gaseous molecules. In contrast, nitrobenzene is a liquid at room temperature, and thus the

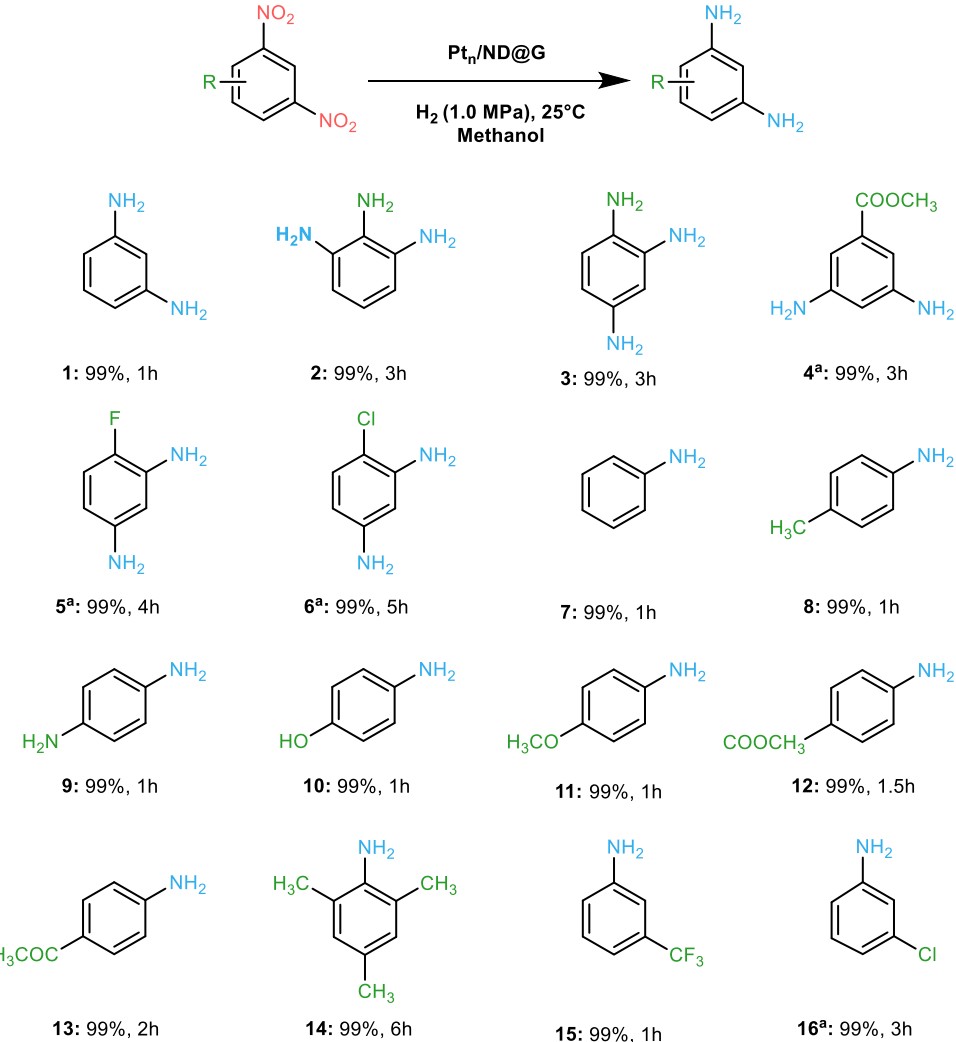

**Fig. 4 | Substrate scope of the hydrogenation reaction of di-nitroarenes and nitroarenes using Pt$_n$/ND@G.** Reaction conditions: 25 °C, 1.0 MPa, 1 mmol of substrate, 10 mL methanol as solvent, 10 mg Pt$_n$/ND@G; >99% conversion was observed; yield was determined by GC. [a]0.5 mmol of substrate.

nitrobenzene vapors can easily diffuse into the IR reaction cell by a high-flow nitrogen purge. For both Pt$_n$/ND@G and Pt$_1$/ND@G, two characteristic peaks were observed at 1525 cm$^{-1}$ and 1348 cm$^{-1}$ in the DRIFTS before the introduction of H$_2$, which were assigned to the asymmetrical stretching vibration ($v_{as}$(NO$_2$)) and symmetrical stretching vibration ($v_s$(NO$_2$)) of the nitro group, respectively[43]. For Pt$_n$/ND@G (Fig. 6a), the band at $v_{as}$(NO$_2$) (1525 cm$^{-1}$) and $v_s$(NO$_2$) (1348 cm$^{-1}$) disappeared rapidly accompanying the inflow of hydrogen; and two fresh bands were observed at 1607 cm$^{-1}$ and 1500 cm$^{-1}$ attributed to the N–H bending vibration of aniline. This dynamic variation confirms the activation and hydrogenation of −NO$_2$ to −NH$_2$ over Pt$_n$/ND@G. In the presence of Pt$_1$/ND@G (Fig. 6b), there were no observed changes in the band density ascribed to nitro groups (1525 cm$^{-1}$ and 1348 cm$^{-1}$) along with flowing H$_2$. Similarly, even after passing H$_2$ for 32 min, no peaks of N–H bending vibrations were detected. Hence, the adsorption of −NO$_2$ on the single Pt site adversely affected the subsequent adsorption and activation of H$_2$, which is consistent with the DFT calculations.

Combining the experimental and theoretical results, we conclude that the multiple active sites in Pt$_n$/ND@G could promote the co-adsorption/activation of substrates and H$_2$, and the release of active sites due to the moderate adsorption ability of intermediates and product, which facilitated the progression of the multi-step hydrogenation reaction.

## Discussion

In summary, we employed fully exposed Pt clusters catalysts (Pt$_n$/ND@G) supported on ND@G by a deposition-precipitation method. With the help of electron microscopy images, XAFS spectra, and DRIFTS spectra, the Pt$_4$ clusters in the Pt$_n$/ND@G were identified. Pt$_n$/ND@G displayed dramatically superior hydrogenation performance (TOF = 40,647 h$^{-1}$, yield >99%) compared to Pt$_1$/ND@G, Pt$_p$/ND@G, or any known catalyst, for the multi-step hydrogenation of 2,4-DNT under room temperature. Besides, the fully exposed Pt$_n$/ND@G demonstrated excellent stability in the recycling tests and exhibited excellent catalytic performance in the multi-step hydrogenation for various di-nitroarenes containing various substituent groups. DFT calculations and absorption experiments confirmed that the fully exposed Pt clusters could simultaneously adsorb substrates and H$_2$ to enhance the catalytic activity. Further calculations reveal that the adsorption of H$_2$ by Pt$_4$@Gr (−0.16 eV) is more favored thermodynamically than that of Pt$_1$@Gr (0.01 eV) after the adsorption of 2,4-DNT. Compared with Pt(111), the moderate adsorption ability for intermediates and products on Pt$_4$@Gr prevented their accumulation on active sites and contributed to its high activity level. Therefore, our work elucidates the advantages of utilizing unique FECCs for promoting the efficient multi-step hydrogenation of di-nitroarenes under mild conditions, while also paving an avenue for the rational design of highly active and

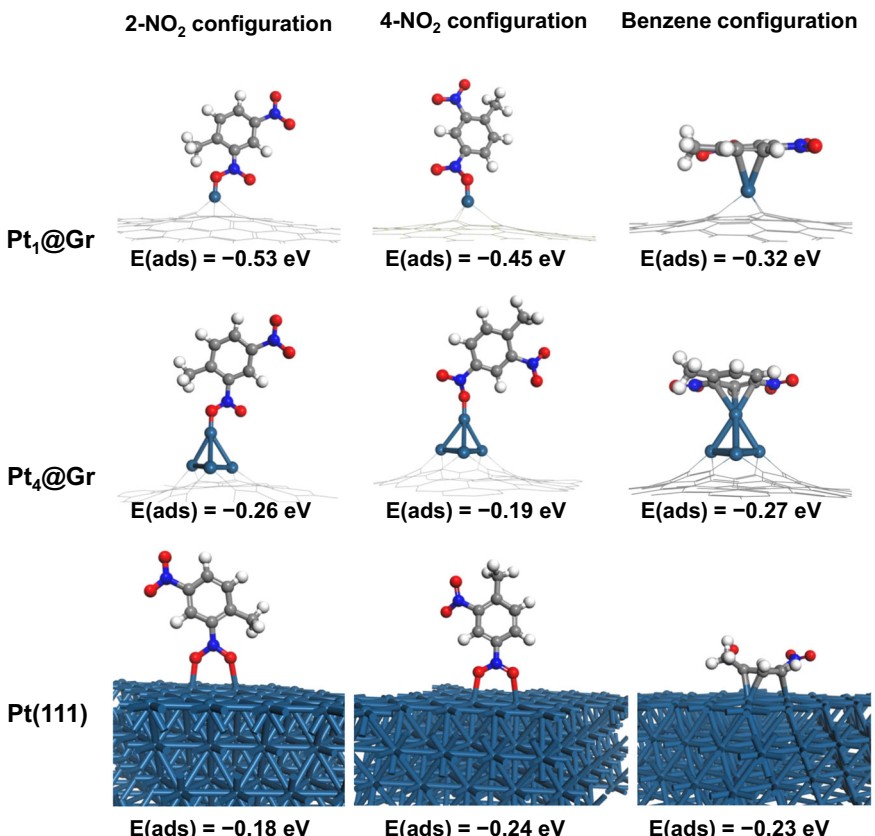

**Fig. 5 | DFT calculations.** Three possible adsorption models (at 2-NO₂- configuration, at 4-NO₂- configuration, and benzene- configuration) and the corresponding adsorption energy of 2,4-DNT over the Pt₁@Gr, Pt₄@Gr, and Pt(111).

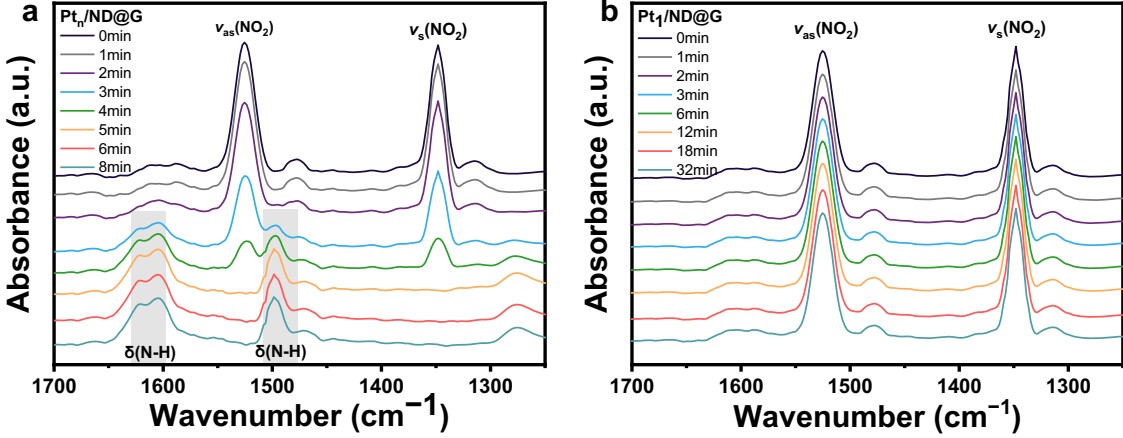

**Fig. 6 | Investigations on reaction mechanism of 2,4-DNT hydrogenation.** In situ DRIFTS spectra of nitrobenzene hydrogenation in the presence of (**a**) Ptₙ/ND@G and (**b**) Pt₁/ND@G by flowing H₂ as a reaction gas.

selective hydrogenation catalysts applied for multi-step hydrogenation reactions.

## Methods

### Materials

Nanodiamond (ND) powders were procured from Beijing Grish Hitech Co., China. The graphene, active carbon, Al₂O₃, TiO₂, and SiO₂ powders were obtained from Aladdin. The commercial catalysts (5 wt% Pt/C) were purchased from Alfa Aesar. Chloroplatinic acid (H₂PtCl₆·6H₂O) and sodium formate ((HCOONa)) were analytical regent and purchased from Sinopharm Co. Ltd.

### Preparation of catalysts

The nanodiamond@graphene (ND@G) was synthesized via thermal annealing, where pristine ND powders were subjected to 1100 °C under an Ar flow for 4 h. The Ptₙ/ND@G, with 0.5 wt% Pt loading, was prepared using the deposition-precipitation method with H₂PtCl₆·6H₂O as the precursor. Initially, ND@G (200 mg) was ultrasonically dispersed in 25 mL deionized water in a 100 mL round-bottom flask for 30 min. Subsequently, the suspension was heated to 100 °C in an oil bath under reflux conditions for another 30 min, followed by the introduction of sodium formate. H₂PtCl₆·6H₂O aqueous solution (equivalent to 0.5 wt% Pt loading on ND@G support) was added into 2 mL deionized water

and ultrasonicated for 5 min. After stirring for an additional 30 min, the above solution was dropwise into the carbon support suspension with magnetic stirring at 100 °C, and the mixture continued to be stirred at 100 °C for 1 h. Once completed, the mixture was cooled to room temperature, filtered, and rinsed multiple times with deionized water. Ultimately, the powders were dried in a vacuum at 60 °C and reduced in $H_2$ at 200 °C. Pt/G (graphene), Pt/ND (nanodiamond), Pt/AC (active carbon), Pt/$Al_2O_3$, Pt/$TiO_2$, and Pt/$SiO_2$ were synthesized by the same method and loading as $Pt_n$/ND@G. The $Pt_1$/ND@G with 0.1 wt% Pt loading was synthesized by an impregnation method. Typically, an amount of aqueous $H_2PtCl_6 \cdot 6H_2O$ solution was added to 2 mL of ethanol. Then, ND@G (200 mg) was dispersed into the solution, and the solution was kept under vigorous stirring. After aging for 20 h, the precipitate was dried overnight at 60 °C in a vacuum. Finally, the powder was treated under the $H_2$ at 200 °C. The $Pt_p$/ND@G with 1.0 wt% Pt loading was prepared by an impregnation method. ND@G (200 mg) was dispersed in 2 mL of deionized water in a beaker, and then certain amounts of $H_2PtCl_6 \cdot 6H_2O$ aqueous solution were added; after the mixture was stirred until dry. After the obtained powder was dried in a vacuum at 60 °C, it was reduced to 200 °C under an atmosphere of $H_2$.

### Characterizations

High-resolution transmission electron microscopy (HRTEM) analysis was performed using FEI Tecnai G2 F20 operating at 200 kV. Aberration-corrected high-angle annular dark-field scanning transmission electron microscopy (AC-HAADF-STEM) images were taken on a JEOL JEM ARM 200CF working at 200 kV. The X-ray diffraction (XRD) patterns were measured with a D/MAX-2500 PC X-ray diffractometer with monochromated Cu Kα radiation (λ = 1.54 Å). The $H_2$-$O_2$ titration measurements were carried out with a Micromeritics AutoChem II 2920 equipped with a thermal conductive detector. X-ray photoelectron spectroscopy (XPS) was conducted on an ESCALAB 250 instrument with an Al Kα X-ray source (1489.6 eV, 150 W, 50.0 eV pass energy). The Pt contents in catalysts were determined by a Profile Spec inductively coupled plasma optical emission spectroscopy (ICP-OES) spectrometer. X-ray absorption structure (XAS) spectra were recorded at the Pt $L_3$-edge on beamline 1W1B at the Beijing Synchrotron Radiation Facility (SSRF, 2.5 GeV with a current of 250 mA), utilizing a Si (111) double crystal monochromator. The XAS data were acquired in transmission mode using ionization chambers as detectors. Spectra were acquired under ambient conditions. The XAS data were processed and analyzed using Demeter software package[44]. Wavelet transform (WT) of Pt $L_3$-edge EXAFS oscillations were implemented based on Morlet wavelets using HAMA Fortran[45,46].

In situ diffuse reflectance infrared Fourier transform infrared spectroscopy (DRIFTS) experiments were conducted on a Thermo Scientific Nicolet IS10 Fourier transform infrared spectrometer equipped with an MCT and a reaction cell. The resolution of the DRIFTS spectrum was 4 cm$^{-1}$ over 32 scans. The detailed pre-treatment and test conditions are given as follows. As for CO-DRIFTS measurements, the sample was diluted with KBr, loaded into the cell, and calcined in $H_2$ at 200 °C for 1 h. After cooling the sample to 30 °C in a high-purity He stream, a background spectrum was collected. Subsequently, the CO/ He (5%/95%) was introduced into the cell to saturate the sample surface with CO, and DRIFTS spectra were collected until the adsorption peak intensity no longer increased. Finally, the gas flow was switched to a pure He stream to collect CO chemisorption spectra. For in situ DRIFTS measurements of nitrobenzene adsorption and surface reaction, the sample diluted with KBr was loaded into the cell and calcined in $H_2$ at 200 °C for 1 h. After cooling the sample to 30 °C in a high-purity He stream, a background spectrum was collected. Nitrobenzene was introduced into the cell for 30 min followed by purging He to remove the physically adsorbed molecule before collecting DRIFTS signals. Finally, the spectra for the hydrogenation process were collected per 1 min after the introduction of $H_2$.

### Reaction evaluation

The hydrogenation of di-nitroarenes was conducted in a high-pressure autoclave with a magnetic stirrer (700 rpm). Typically, a 10 mL mixture of di-nitroaromatics reactants, catalysts, solvent, and internal standard dodecane was loaded into the reactor. After sealing, the autoclave was flushed with Ar for 3 times and $H_2$ for 3 times, and then it was heated to a given temperature with a magnetic stirrer. Once the temperature was reached, $H_2$ was introduced and kept at a desired pressure in the reactor to initiate the reaction. Upon completion of the reaction, the autoclave was cooled to room temperature, and the remaining hydrogen gas was discharged. The product was analyzed by gas chromatography.

### Computational details

DFT calculations were performed by Vienna ab initio simulation package (VASP)[47,48]. The interaction between ion and electron is described by the projector augmented wave (PAW)[49,50] potential. The electron exchange and correlation energies are treated by Perdew–Burke–Ernzerhof (PBE) funcitional[51]. The van der Waals dispersion corrections are by the late parameter (D3)[52,53]. A cutoff energy is set by 500 eV, the convergence tolerance is $10^{-5}$ eV and 0.03 eV/Å for electronic and ionic optimizations, respectively. The pt1@Gr model is constructed by placing a metal atom at a carbon defect of a $p(6 \times 6)$ graphene layer. Also, the adject three carbon atoms are removed from the graphene layer, forming a defected site in which a $Pt_4$ cluster is located. A $p(6 \times 6)$ Pt(111) surface with four layers is used, which has the bottom two layers fixed. The vacuum layers are set by 20 Å. A $1 \times 1 \times 1$ Gamma-centered Monkhorst–Pack k-point grid was used for sampling the Brillouin zone[54]. The Gaussian smearing method was used with a smearing width of 0.05 eV.

### Reporting summary

Further information on research design is available in the Nature Portfolio Reporting Summary linked to this article.

## Data availability

The data supporting this article and other findings are available from the corresponding authors upon request. Source data are provided with this paper.

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

## Acknowledgements

This work was supported by the National Key R&D Program of China (2022YFA1504500, 2022YFB4003100, 2021YFA1502802), the National Natural Science Foundation of China (22072162, 2202213, 92145301, U21B2092, 21961160722, 91845201), the International Partnership Program of Chinese Academy of Sciences (172GJHZ2022028MI), Shenyang Young Talents Program (RC210435), Dalian National Lab for Clean Energy (DNL Cooperation Fund 202001) and China Petroleum & Chemical Corporation (No. 420043-2). The XAS experiments were conducted in Beijing Synchrotron Radiation Facility (BSRF) and Shanghai Synchrotron Radiation Facility (SSRF). D.M. acknowledges support from the Tencent Foundation through the XPLORER PRIZE. We very much thank Dr. Pengju Ren from Synfuels China Co., Ltd. for his great help in the discussion of the DFT calculations and reaction mechanism.

## Author contributions

H.L. and D.M. conceived the project. Y.S. conducted material synthesis and carried out the catalytic performance test. J.D. conducted the HRTEM, STEM characterizations and analyses. M.W. and J.C. conducted the X-ray absorption fine structure spectroscopic measurements and analyzed the data. S.X. and N.W. contributed to the AC-HAADF-STEM characterization. Y.J. and X.W. performed the DFT calculations. X.C. and Y.W. performed some of the synthesis experiments. The manuscript was primarily written by Y.S., Y.J., D.X., H.L. and D.M. All authors contributed to discussions and manuscript review.

## Competing interests

The authors declare no competing interests.
