## [Peer Review File · Nature Communications]

Fully Exposed Pt Clusters for Efficient Catalysis of Multi-Step Hydrogenation ReactionsREVIEWER COMMENTS

Reviewer #1 (Remarks to the Author):

This work reports a study on the hydrogenation activity of highly dispersed Pt catalyst on carbon. Such systems are extremely challenging to characterize, rendering their modeling very risky and sometime hazardous. I do not question the performance of the catalyst prepared on carbon, but several elements leave me cautious about the interpretation of the results obtained. In fact, major revisions are necessary before this article can be considered for publication. The mechanistic investigation in the manuscript is of moderate interest, and the presentation/discussion of the experimental/modeling data are too primary.

- 1) The term graphene (ND@G) is not appropriate for this sp³/sp² structure with pronounced curvature, please consider a modification.
- 2) It is said in the text that "The synthetic method of ND@G was based on our previous work, and the detailed preparation processes can be found in the Support Information". This is not the case.
- 3) The characterizations presented, and in particular the STEM and DRIFT analyses, suggest a mixture of clusters and single atoms rather than the proposed Pt₄ model. Cooperative catalysis between these species using hydrogen spillover has been reported in the literature (ref 16). The hydrogen spillover must be evaluated as well as the possibility of such cooperativity (by calculations for example).
- 4) The proposed coordination for the isolated Pt atom and the clusters involves a monovacancy. Such highly reactive species are not expected to survive the conditions used for catalyst preparation. This model does not seem justified to me, unless the authors can demonstrate their existence on their carbon material (for example by EPR). Furthermore, for Pt₁@C the CN does not correspond to the model presented and the presence of chlorine (CN=2) does not appear in the model. The proposed models seem really oversimplified to me to be used for rationalization.
- 5) The observed solvent effect may also be linked to a greater hydrogen spillover; It is well known that oxygenated molecules can serve as a shuttle for H species originating from the spillover. This must be considered and discussed in the text.
- 6) Catalyst recycling tests should not be performed at 100% conversion; this does not allow us to conclude as to the possible stability of the catalyst.
- 7) The final part of the article aimed at explaining the difference in performance between isolated atoms and clusters is of little interest. It is well known in the literature that isolated atoms do not perform well for these reactions, and this is discussed in the introduction to the article. On the other hand, it would make sense to try to explain the difference in TOF observed between the clusters and the Pt particles. It is on this last point that the authors must concentrate.

Reviewer #2 (Remarks to the Author):

In this manuscript, the authors fabricated a fully exposed-Pt-cluster catalyst, consisting of an average of four Pt atoms supported on a defective nano-diamond graphene (Ptn/ND@G), for the complex multi-step hydrogenation reactions of di-nitroaromatics. The physicochemical properties of the catalysts were thoroughly investigated using HAADF-STEM, XAFS, DRIFTS and DFT, which provided compelling support for the proposed reaction mechanism. The robust catalytic performance and good stability for the hydrogenation of 2,4-dinitrotoluene under mild reaction conditions outperform other known catalysts. This work offers a new application for the multi-step hydrogenation into the fine chemical sector under mild conditions. Overall, the manuscript reports a nice and systematic work, with a logical methodological approach and satisfactory discussion in the context of literature data. I recommend acceptance of this manuscript in Nature Communications after minor revision, comments are shown below:

1. The nanodiamond@graphene hybrid carbon material (ND@G) is an intriguing system. A brief discussion explaining why the ND@G support could highly disperse the metal atoms would be helpful.

2. The conclusion that the cluster is only single-atomic-layer thick, derived from the STEM investigation (Fig. 1e). However, the model of the Pt₄ cluster is standing upright and would be higher than a monolayer. Is this in agreement with the experimental observations?
3. The author should carefully check the figure legends of Fig. 2g, and all other figures should be double-checked.
4. In Fig. 3, the details for the amount of Pt to substrate and reaction conditions should be explicitly provided in the figure captions to allow the reader to qualify the results.
5. The paper states that the fully exposed Pt clusters catalyst provide excellent reactivity compared with other catalytic systems. However, no data is presented on a commercial catalyst. The authors should collect their own data under same experimental conditions as a control.
6. The authors' experimental results demonstrate that the performance of Pt_n/ND@G is stable in a 5-cycle test with no significant loss of activity. I wonder whether the catalyst has any leaching, ICP of the hot-filtration experiment should be provided.
7. The author should include more comments (in the SI) on side products for low-selectivity catalysts (e.g., Pt/G or Pt/ND), which is helpful for understanding different mechanism between these catalysts.

Reviewer #3 (Remarks to the Author):

The presented paper focuses on the hydrogenation of nitro-aromatics catalyzed by Pt and highlights the high activity of small clusters compared to single-atom and nanoparticles. The manuscript is well-organized and easy to follow. The written expression is outstanding; only a misspelt in line 167 was found "beclearly", i.e., lack of space. Some statements need to be clarified before publication. The repeated terminology "fully exposed" is not defined and is rather unclear and unprecise. These tiny clusters seem to have a flat morphology parallel to the support (line 105); therefore, half of the cluster is exposed to the reaction medium. What does "both" refer to in line 64? Some references are missing, for instance, backing the sentence in line 72. The activity is described in lines 113 and 114, but for which process? In lines 146-147, the lack of Pt diffraction peaks is associated with high dispersion, but it is actually due to the small size of these clusters. A more critical issue is the lack of clarity in the neighbouring atoms to Pt, i.e. C or O. According to the positive oxidation state, Pt could be surrounded by oxygen; however, the argumentation is not pursued. Catalysts in line 231 are required to be accurately characterized or provide a reference to their characterization. In the DFT model, Pt presents negative charges which lack units. At the end of line 299, "electronic structure" should be Bader analysis; there is a considerable difference between these terms. The atomistic models are inconclusive: H₂ should dissociate before reacting, and therefore, the co-adsorption energies are practically unrelated to the hydrogenation activity. Does the simulation consider long-range dispersion corrections?

REVIEWER COMMENTS

Reviewer #1 (Remarks to the Author):

This work reports a study on the hydrogenation activity of highly dispersed Pt catalyst on carbon. Such systems are extremely challenging to characterize, rendering their modeling very risky and sometime hazardous. I do not question the performance of the catalyst prepared on carbon, but several elements leave me cautious about the interpretation of the results obtained. In fact, major revisions are necessary before this article can be considered for publication. The mechanistic investigation in the manuscript is of moderate interest, and the presentation/discussion of the experimental/modeling data are too primary.

Comments 1. The term graphene (ND@G) is not appropriate for this sp^3/sp^2 structure with pronounced curvature, please consider a modification.

Response: We thank the reviewer for this valuable suggestion. The nanodiamond@graphene hybrid carbon material (ND@G) is an intriguing support that our team has studied for many years. We can systematically control the microstructure and surface chemistry of nanodiamond (ND) through annealing treatments in an inert atmosphere. After a thermal treatment at 1100°C, the surface of ND can be reconstructed into an ultrathin, curved, and defect-rich sp^2 graphene nano-shell reinforced by a sp^3 diamond nano-core (Figure R1) that could anchor isolated metal atoms through Metal–C bond to obtain atomically dispersed catalysts. The role of nanodiamond is that it is a perfect substrate for the grown of highly defective few-layer

graphene outersheet with curvature, which is pivotal to preparation of Pt catalysts. Therefore, for this sp^3/sp^2 structure with pronounced curvature, we denoted it as ND@G. And the relevant details of this interesting material could be followed in our previous publications. (J. Am. Chem. Soc. 2018, 140, 13142–13146, ACS Catal. 2019, 9, 5998–6005, Nat. Commun. 2023, 14, 2588.)

Figure R1. (a) HRTEM image of the as-prepared ND@G support with nanodiamond core and defective graphene shell; (b) Structure diagram of ND@G.

Comments 2. It is said in the text that "The synthetic method of ND@G was based on our previous work, and the detailed preparation processes can be found in the Support Information". This is not the case.

Response: We thank the reviewer for the nice suggestion. We're terribly sorry for the error of expression. Indeed, the detailed preparation processes can be found in the Methods part. So, we have made correction in the revised manuscript according to the reviewer's comments. "The synthetic method of ND@G was based on our previous work, and the detailed preparation processes can be found in the Methods." (Please see Line 129-130)

Comments 3. The characterizations presented, and in particular the STEM and DRIFT analyses, suggest a mixture of clusters and single atoms rather than the proposed Pt₄ model. Cooperative catalysis between these species using hydrogen spillover has been reported in the literature (ref 16). The hydrogen spillover must be evaluated as well as the possibility of such cooperativity (by calculations for example).

Response: We thank the reviewer for pointing this out for us. Indeed, because of some uncontrollable factors in the preparation process, it is inevitable that there are a few of Pt single atoms in the Pt cluster catalysts. However, most of Pt species are still in the form of clusters. Following the reviewer's suggestion, we carefully considered the possibility of the synergistic effects of cluster and single-atom, and calculated H-transfer from cluster sites to single-atom sites via two H-spillover routes:

- (1) H-transfer by carbon defect sites (Table R1). Three carbon defect sites are constructed, which respectively missed 1, 2, 3 carbon atoms, named as NGD@GV₁, NGD@GV₂, and NGD@GV₃. And the results show that i) the adsorption of H at NGD@GV₁ and NGD@GV₃ site is too strong (-2.23 and -2.72 eV) to desorb H; (ii) the adsorption of H at NGD@GV₂ is thermodynamically unfavored (0.15 eV); (iii) the H transfer from (0.67 eV).
- (2) H-transfer by CH₃OH (Table R2). The calculations show that CH₃OH--H intermediate cannot exist stably as a H-carrier because it can easily dissociate into CH₃ and H₂O. And the same results have been reported in the literature (Nat. Commun. 2022, 13, 1457).

As a result, we think that there is probably no H-transfer from Pt₄ clusters to Pt single atoms in the present catalytic system.

Table R1. The adsorption energies (E_{ads}) of H* on Pt₄@GV₃, NGD@GV₁, NGD@GV₂, NGD@GV₃, and Pt₁@GV₁ sites, respectively.

	$E_{\text{ads}}(\text{eV})$
H*(at Pt ₄ site)	-0.22
H*(at C near Pt ₄ site)	0.67
H*(at C far from Pt ₄ site)	1.34
H* (at NGD@GV ₁)	-2.23
H*(at NGD@GV ₂)	0.15
H* (at NGD@GV ₃)	-2.72
H*(at C far from Pt SP site)	0.86
H*(at C near Pt SP site)	0.34
H*(at Pt SP site)	-0.52

Table R2. The H-transfer by CH₃OH from Pt₄@GV₃ site to Pt₁@GV₁ site.

	$E_{\text{r}}(\text{eV})$
CH ₃ OH+H*(at Pt ₄ site)	0.00
CH ₃ OH*+H*(at Pt ₄ site)	-0.22
CH ₃ OH--H*(at Pt ₄ site)	0.04
CH ₃ OH--H= CH ₃ +H ₂ O	1.23
CH ₃ OH*+H*(at Pt site)	-0.83
CH ₃ OH--H*(at Pt site)	-0.43
CH ₃ OH+H*(at Pt site)	-0.30

Comments 4. The proposed coordination for the isolated Pt atom and the clusters

involves a monovacancy. Such highly reactive species are not expected to survive the conditions used for catalyst preparation. This model does not seem justified to me, unless the authors can demonstrate their existence on their carbon material (for example by EPR). Furthermore, for Pt₁@C the CN does not correspond to the model presented and the presence of chlorine (CN=2) does not appear in the model. The proposed models seem really oversimplified to me to be used for rationalization.

Response: We thank the reviewer for this nice comment.

(1) To address this concern, we performed Raman spectroscopy (Figure R2) to further characterize the degree of defect on ND@G, Pt₁/ND@G, and Pt_n/ND@G. The D band (-1400 cm⁻¹) originates from the disordered carbon, whereas the G band (-1580 cm⁻¹) is related to the vibration of sp² carbon atoms. The intensity ratio of D band to G band (I_D/I_G) indicates the degree of disorder and defects of carbon materials. These results indicate that all the ND@G, Pt₁/ND@G, and Pt_n/ND@G have rich carbon defective sites. In addition, the literature has also proved that the monovacancy can exist stably and serve as an adapted host for trapping metal atoms. (J. Am. Chem. Soc. 2022, 144, 2171-2178.)

(2) In fact, the effect of a small amount of Cl coordination on the Pt single atom sites was ignored in the construction of the DFT model (ACS Catal. 2022, 12, 8104–8115) and we drew on this idea when building the model in the original manuscript. Herein, we constructed the Pt₁Cl₂@GV₂ model according to the coordination structure from experimental results (CN(Pt-Cl) = 2.1, CN(Pt-C) = 1.7), and calculated the adsorption energies of the intermediates in the hydrogenation of 2,4-DNT to make sure

the reliability of our conclusion. As shown in Table R3, due to the steric hindrance on single Pt atom, the very weak adsorption of reactants (-0.09 eV) and intermediates (-0.09, -0.13 and -0.22 eV) would occur on the $\text{Pt}_1\text{Cl}_2@\text{GV}_2$. As a contrast, the adsorption of 2,4-DNT on $\text{Pt}_1@\text{GV}_2$ is stronger than on Pt_4 clusters and $\text{Pt}(111)$ surface (-0.45 vs. -0.27 and -0.23 eV), consistent with our previous discussion on $\text{Pt}_1@\text{GV}_1$. Therefore, although Pt-Cl bond is detected in our single-atom catalyst, the working active site is Pt metal atom only coordinated with C atom. Our models of $\text{Pt}_1@\text{GV}_2$ and $\text{Pt}_1@\text{GV}_1$ are also reasonable for explaining the experiment.

Figure R2. Raman spectra of ND@G, $\text{Pt}_1/\text{ND@G}$, and $\text{Pt}_n/\text{ND@G}$.

Table R3. The adsorption energies E_{ads} (eV) of intermediates on different single Pt atom models.

	Pt ₁ @GV ₂	Pt ₁ Cl ₂ @GV ₂	Pt ₁ @GV ₁
2,4-DNT	-0.45 eV	-0.09 eV	-0.53 eV
2N4AT	-0.41 eV	-0.09 eV	-0.63 eV
2A4NT	-0.54 eV	-0.13 eV	-0.67 eV
2,4-DAT	-0.83 eV	-0.22 eV	-0.84 eV

Comments 5. The observed solvent effect may also be linked to a greater hydrogen spillover; It is well known that oxygenated molecules can serve as a shuttle for H species originating from the spillover. This must be considered and discussed in the text.

Response: We thank for the reviewer's nice suggestion. We agree with the reviewer. It has been reported that the oxygenate molecules containing a carbonyl functional group (such as aldehydes, ketones, and esters) can serve as a shuttle for H species originating from the spillover. But H spillover through the methanol is hard (Nat. Commun. 2022, 13, 1457). Similarly, our calculations showed that it is difficult for H-spillover from Pt₄ clusters to Pt single atoms by CH₃OH. And our evidence can be found in the responses to Comment 3.

Comments 6. Catalyst recycling tests should not be performed at 100% conversion; this does not allow us to conclude as to the possible stability of the catalyst.

Response: We thank the reviewer for this constructive suggestion. We add 5 recycling tests at low conversion (Figure R3). The HAADF-STEM and ICP analyses also applied

to characterize the used catalyst after the 5th cycle. The results and discussion were shown in the revised manuscript. “The Pt_n/ND@G can be used for 5 cycles at low conversion without any loss in activity (Supplementary Fig. 8). The STEM images and ICP results of Pt_n/ND@G after the reaction indicated the absence of any aggregation and leaching, demonstrating its outstanding structural stability during the reaction (Supplementary Fig. 9 and Supplementary Table 4).” (Please see Line 267-271)

Figure R3. Recycling test for 2,4-DNT hydrogenation over the Pt_n/ND@G.

Comments 7. The final part of the article aimed at explaining the difference in performance between isolated atoms and clusters is of little interest. It is well known in the literature that isolated atoms do not perform well for these reactions, and this is discussed in the introduction to the article. On the other hand, it would make sense to try to explain the difference in TOF observed between the clusters and the Pt particles.

It is on this last point that the authors must concentrate.

Response: We thank the reviewer for the nice suggestion. Following the reviewer's suggestion, we add the DFT calculations and discuss the difference of catalytic performance between the clusters and the Pt particles. Although the easier H₂ dissociation is found on Pt(111) (Table R4), the adsorption of the product 2,4-DAT are much stronger than Pt₄ clusters (Table R5). Consequently, the product would preferentially adsorb and accumulate on Pt(111), leading to a declined activity. Above results and discussion are shown in the revised manuscript. We add and rewrite "The hydrogenation of 2,4-DNT is a super-exothermic reaction, which is verified experimentally and theoretically. And our DFT calculations also show the same result (Supplementary Table 5). This leads to the fact that once the reactants can be adsorbed onto the active site, it will be a key factor for the reaction activity that whether the product and/or intermediates can desorb and release the active site successfully. To understand the difference for the hydrogenation on Pt single atoms, Pt clusters, and Pt metal particles. Based on the results of XAFS and STEM, we used Pt₁@Gr, Pt₄@Gr and Pt(111) (Supplementary Fig. 10) as the models, respectively." (Please see Line 302-309); "On Pt(111) surface, the adsorption by 2-NO₂⁻, 4-NO₂⁻, and benzene- has the adsorption energies of -0.18, -0.24, and -0.23 eV, respectively." (Please see Line 315-317); "On Pt(111) surface, the dissociation of H₂ is spontaneous, which is exothermic by 1.13 eV, implying that a potentially high catalytic activity of 2,4-DNT." (Please see Line 333-335); "Compared Pt(111) with Pt₄@Gr, it is found that although easier H₂ dissociation (-1.13 eV) on Pt(111), the adsorption of the product 2,4-DAT is too strong

(Supplementary Table 7) to desorb, which makes the product accumulation on active sites and causes a decrease in activity.” (Please see Line 339-342)

Table R4. The calculated reaction energies ($E1$, $E2$, $E3$ unit: eV) of $R1$, $R2$, $R3$ on Pt(111), Pt₄@Gr, Pt₁@Gr catalysts with PBE+ZPE and PBE+D3+ZPE. $R1$, $R2$, and $R3$ is respectively represented the reaction steps of 2,4-DNT = 2,4-DNT*, H₂+2,4-DNT* = H₂*+2,4-DNT* and H₂*+2,4-DNT* = 2H*+2,4-DNT*.

	PBE+ZPE			PBE+D3+ZPE		
	$E1$	$E2$	$E3$	$E1$	$E2$	$E3$
Pt(111)	-0.23 eV	/	-1.13 eV	-1.41 eV	/	-1.30 eV
Pt ₄ @Gr	-0.27 eV	-0.16 eV	-0.87 eV	-1.39 eV	-0.10 eV	-0.74 eV
Pt@Gr	-0.53 eV	0.01 eV	0.18 eV	-1.71 eV	-0.28 eV	0.47 eV

Table R5. The adsorption energies E_{ads} (eV) of intermediates on different catalysts.

	Pt(111)	Pt ₄ @Gr	Pt ₁ @Gr
2,4-DNT	-0.23 eV	-0.27 eV	-0.53 eV
2N4AT	-0.73 eV	-0.44 eV	-0.63 eV
2A4NT	-0.80 eV	-0.45 eV	-0.67 eV
2,4-DAT	-1.57 eV	-0.74 eV	-0.84 eV

Reviewer #2 (Remarks to the Author):

In this manuscript, the authors fabricated a fully exposed-Pt-cluster catalyst, consisting of an average of four Pt atoms supported on a defective nano-diamond graphene (Pt_n/ND@G), for the complex multi-step hydrogenation reactions of di-nitroaromatics. The physicochemical properties of the catalysts were thoroughly investigated using HAADF-STEM, XAFS, DRIFTS and DFT, which provided compelling support for the proposed reaction mechanism. The robust catalytic performance and good stability for

the hydrogenation of 2,4-dinitrotoluene under mild reaction conditions outperform other known catalysts. This work offers a new application for the multi-step hydrogenation into the fine chemical sector under mild conditions. Overall, the manuscript reports a nice and systematic work, with a logical methodological approach and satisfactory discussion in the context of literature data. I recommend acceptance of this manuscript in Nature Communications after minor revision, comments are shown below:

Comments 1. The nanodiamond@graphene hybrid carbon material (ND@G) is an intriguing system. A brief discussion explaining why the ND@G support could highly disperse the metal atoms would be helpful.

Response: We thank the reviewer for this nice comment. Strong electron transfer between Pt and the rich structural defects in the graphene shell of ND@G results in strong metal–support interactions (SMSIs). These interactions are energetically favorable for anchoring metal atoms and stabilizing atomically dispersed metal species through metal-C bonding.

Comments 2. The conclusion that the cluster is only single-atomic-layer thick, derived from the STEM investigation (Fig. 1e). However, the model of the Pt₄ cluster is standing upright and would be higher than a monolayer. Is this in agreement with the experimental observations?

Response: We thank the reviewer for the nice suggestion. We are very sorry for not

explaining this question clearly. The claim of monolayered Pt clusters from the STEM investigation means that there are no Pt atoms covering each other and all Pt atoms in the clusters can be exposed to reactants during reaction. As illustrated in the top and side views of the DFT-calculated model in Figure R4 (a) and (b), the computed Pt₄ cluster has no Pt-Pt overlay in the thickness direction and thus contains single Pt layer.

Figure R4. Top (a) and side (b) views of the DFT-computed Pt₄ cluster model, respectively. Gray balls for carbon while blue balls for platinum.

Comments 3. The author should carefully check the figure legends of Fig. 2g, and all other figures should be double-checked.

Response: We thank the reviewer for the nice comment. Based on your valuable comments, we have carefully checked and corrected the figure legends of Fig. 2g and all other figures.

Comments 4. In Fig. 3, the details for the amount of Pt to substrate and reaction conditions should be explicitly provided in the figure captions to allow the reader to

qualify the results.

Response: We thank the reviewer for this nice comment. We have added an explicit description about the reaction conditions and the amount of Pt to substrate in the revised manuscript. “Fig. 3 (a) The possible reaction pathways for the hydrogenation of 2,4-DNT; (b) Catalytic results on Pt_n/ND@G in various solvents. Reaction conditions: 5 mg of catalyst, 1 mmol of 2,4-DNT, 10 mL of solvent, 1 MPa of H₂, 60 °C, and reaction time 20min. (c) Catalytic performance of Pt loaded on the series of supports. Reaction conditions: the amount of Pt to the substrate is 0.0256 mol%, 1 mmol of 2,4-DNT, 10 mL of methanol, 1 MPa of H₂, 25 °C, and reaction time 2h. (d) TOF values of ND@G, Pt₁/ND@G, Pt_n/ND@G, and Pt_p/ND@G. Reaction conditions: the amount of Pt to the substrate is 0.0256 mol%, 1 mmol of 2,4-DNT, 10 mL of methanol, 1 MPa of H₂, 25 °C, and reaction time 2h.” (Please see Line 273-281)

Comments 5. The paper states that the fully exposed Pt clusters catalyst provide excellent reactivity compared with other catalytic systems. However, no data is presented on a commercial catalyst. The authors should collect their own data under same experimental conditions as a control.

Response: We thank the reviewer for pointing this out for us. Following the suggestion, we have added the experimental results and corresponding description of commercial Pt/C catalyst under same experimental conditions as a control in the revised manuscript. The results show that the fully exposed Pt cluster catalyst provides the excellent reactivity compared with other catalytic systems and commercial Pt/C catalyst. We add “Furthermore, the commercial 5 wt% Pt/C (characterized by STEM, Supplementary Fig. 4) were used as reference samples. As shown in Figure 3c, compare with Pt_n/ND@G, the commercial Pt/C catalyst displayed low activity.” (Please see Line 238-

240)

Comments 6. The authors' experimental results demonstrate that the performance of Pt_n/ND@G is stable in a 5-cycle test with no significant loss of activity. I wonder whether the catalyst has any leaching, ICP of the hot-filtration experiment should be provided.

Response: We thank a lot for your careful review. In the revised manuscript, we add the experiments and characterization of 5 recycling tests at low conversion (Our answer to the Reviewer #1, Comments 6). We add “The Pt_n/ND@G could be used for 5 cycles at low conversion without any loss in activity as shown in Supplementary Fig. 8. The STEM images and ICP results of Pt_n/ND@G after the reaction indicated the absence of any aggregation and leaching, demonstrating its outstanding structural stability during the reaction (Supplementary Fig. 9 and Supplementary Table 4).” (Please see Line 267-271)

Comments 7. The author should include more comments (in the SI) on side products for low-selectivity catalysts (e.g., Pt/G or Pt/ND), which is helpful for understanding different mechanism between these catalysts.

Response: We appreciate the reviewer for the suggestion. In Figure R5, we showed the possible reaction pathways for the multi-step hydrogenation of 2,4-DNT, elucidating the conversion of 2,4-DNT into 2,4-DAT through two parallel pathways involving consecutive reaction steps. In our work, the reactants (2,4-DNT), products (2,4-DAT)

and primary intermediate products (2A4NT, 2N4AT) were analyzed by GC (Agilent 7890A) equipped with a flame ionization detector (FID). For the catalysts (e.g., Pt/G or Pt/ND), the more amount of semi-hydrogenated product (2A4NT, 2N4AT) was found in the GC results, demonstrating the low selectivity towards 2,4-DAT.

Figure R5. The possible reaction pathways for the hydrogenation of 2,4-DNT.

Reviewer #3 (Remarks to the Author):

The presented paper focuses on the hydrogenation of nitro-aromatics catalyzed by Pt and highlights the high activity of small clusters compared to single-atom and nanoparticles. The manuscript is well-organized and easy to follow.

Comments 1. The written expression is outstanding; only a misspelt in line 167 was

found “beclearly”, i.e., lack of space. Some statements need to be clarified before publication.

Response: We wholeheartedly thank the reviewers for spotting the mistakes in our original submission. In the revised manuscript, we have carefully checked the text and corrected the typos. (Please see Line 173)

Comments 2. The repeated terminology “fully exposed” is not defined and is rather unclear and unprecise. These tiny clusters seem to have a flat morphology parallel to the support (line 105); therefore, half of the cluster is exposed to the reaction medium.

Response: We thank the reviewer for the comment which inspired us to add clearer and more precise discussion of the “fully exposed”. “Fully exposed” means that all metal atoms on the catalyst are fully exposed on the surface, and there is no bulk phase structure. The fully exposed structure well-guarantee the maximum atomic utilization during the reaction. Hence, the fully exposed cluster catalyst can not only provide the multiple metal sites but also maintain a full atomic utilization efficiency. FECC is so highly dispersed that all the metal atoms within it are available for the adsorption and transformation of reactants. Following the reviewer’s advice, we added more detailed discussion in the revised manuscript. “As a cross-dimensional extension to the concept of SACs, fully exposed cluster catalysts (FECCs) emerge as a type of catalyst which all metal atoms on the catalyst are fully exposed on the surface of the active sites without bulk phase structure. This design guarantees full atomic utilization during the reaction, maintaining a full atomic utilization efficiency and providing catalytic sites with

multiple metal atoms. FECC is so highly dispersed, allowing all the metal atoms to be available for the adsorption and transformation of reactants. More importantly, supported FECCs exhibit two distinct features. The primary advantage lies in its ultrasmall size (normally below 1 nm), which eliminates the presence of undesired bulk atoms and reduces the average coordination number of the metal atoms. Another advantage is a small contact angle between the metal and support, forming a layered structure, which enhances the interaction between metal atoms and support, and ultimately increasing the stability of the clusters. From this viewpoint, FECCs with multiple active sites have distinct advantages in overcoming the limitations of SACs for a multi-step reaction.” (Please see Line 94-170) And the detailed description of fully exposed clusters could be followed in our review article (ACS Cent. Sci. 2021, 7, 262–273).

Comments 3. What does “both” refer to in line 64?

Response: We appreciate the reviewer for his/her carefulness. In this sentence, the word “both reacting species” refer to H₂ and 2,4-DNT. We are sorry that our ambiguous writing brings much confusion to the reviewer. Here we added and modified the sentence with reviewers' comments. “Density functional theory (DFT) calculations and absorption experiments reveal that the fully exposed Pt clusters with multiple metallic active sites benefit sufficient sites for co-adsorption reactants and H₂, the subsequent dissociation of H₂, and the release of active sites due to moderate adsorption ability of intermediates and products, leading to enhanced catalytic performance.” (Please see

Line 56-61)

Comments 4. Some references are missing, for instance, backing the sentence in line 72.

Response: We thank the reviewer for this constructive suggestion. Following the suggestion of the reviewer, we add relevant reference. (Reference 4) (Please see Line 65-67)

Comments 5. The activity is described in lines 113 and 114, but for which process?

Response: We thank the reviewer for this constructive suggestion. The activity is described for the hydrogenation of 2,4-DNT. We added this to the revised manuscript “Pt_n/ND@G showed an excellent catalytic activity for 2,4-DNT hydrogenation under room temperature: high TOF (40647 h⁻¹), high yield (>99%), and good stability (5 cycles), which is better than Pt single atoms catalysts (Pt₁/ND@G), Pt nanoparticles catalysts (Pt_p/ND@G) supported over ND@G, or all other known catalysts” (Please see line 114-118)

Comments 6. In lines 146-147, the lack of Pt diffraction peaks is associated with high dispersion, but it is actually due to the small size of these clusters.

Response: We thank the reviewer for this constructive suggestion. we agree with the reviewer that the lack of Pt diffraction peaks is associated with the small size of Pt clusters. We modified the corresponding description in the revised manuscript. “The

Pt_n/ND@G samples did not display diffraction peaks corresponding to Pt crystal, indicating the small size of Pt species.” (Please see Line 149-150)

Comments 7. A more critical issue is the lack of clarity in the neighbouring atoms to Pt, i.e. C or O. According to the positive oxidation state, Pt could be surrounded by oxygen; however, the argumentation is not pursued.

Response: We thank the reviewer for this constructive suggestion. Our catalyst is inevitably exposed to the ambient (moisture and O₂) before the batch experiments and the various characterization techniques. The as-prepared catalysts were reduced in H₂ for 1h before the characterization and all samples were sealed within He-filled containers as soon as synthesized to preserve their intrinsic structures before characterization. Similarly, the oxygen atoms that may be attached to Pt can be removed by hydrogen molecules at the initial stage of the hydrogenation reactions. Therefore, it is unlikely that Pt on the catalyst will be surrounded by oxygen in the hydrogenation experiments and the characterizations.

Comments 8. Catalysts in line 231 are required to be accurately characterized or provide a reference to their characterization.

Response: We thank the reviewer for this constructive suggestion. Following the suggestion, we have added the STEM characterization of these catalysts in the revised manuscript (Please see Line 228-231) and SI (Supplementary Fig. 3).

Comments 9. In the DFT model, Pt presents negative charges which lack units. At the

end of line 299, “electronic structure” should be Bader analysis; there is a considerable difference between these terms.

Response: We thank the reviewer for this constructive suggestion. We add the units of Pt presents negative charges in the DFT model in the revised SI. (Supplementary Fig. 10) We correct the wrong terminology “electronic structure” to “Bader analysis” in the revised manuscript (Please see Line 318)

Comments 10. The atomistic models are inconclusive: H₂ should dissociate before reacting, and therefore, the co-adsorption energies are practically unrelated to the hydrogenation activity. Does the simulation consider long-range dispersion corrections?

Response: We thank the reviewer for this constructive suggestion.

(1) In this liquid phase hydrogenation reaction, typically, the mixture of nitroaromatics reactants, catalysts, solvent and internal standard, was put into the reactor. After being sealed, the autoclave was flushed with Ar for 3 times and H₂ for 3 times, and then it was heated to a given temperature with a magnetic stirrer. After the temperature was reached, hydrogen was introduced and kept at a desired pressure in the reactor to initiate the reaction. And in this reaction, the amount of Pt to the substrate is 0.0256 mol%, which means that the active site is saturated with adsorbed substrates first, and thus H₂ does not adsorb and dissociate before the reaction. Similarly, it has been reported in the literature that the hydrogenation reactions of nitroaromatics reactant also start from the adsorption of the nitrobenzene reactant. (Nat. Commun. 12, 3181 (2023); Angew. Chem. Int. Ed. 62, e202307853 (2023).)

(2) The results included van der Waals corrections (PBE+D3+ZPE) (Table R6) are used to evaluate long-range dispersion corrections. It is found that i) the adsorption energies ($E1$) of 2,4-DNT are over-corrected (-0.23 vs. -1.41 eV; -0.27 vs. -1.39 eV; -0.53 vs. -1.71 eV) on Pt(111), Pt₄ cluster and Pt single atoms; ii) difficult dissociation of H₂ (0.47 eV) on Pt single atoms after adsorbing 2,4-DNT, which causing the low catalytic activity. Therefore, the same trends and conclusions are obtained for PBE+D3+ZPE and PBE+ZPE. Due to the lack of accurate experimental values as a reliable reference, the current method (PBE+ZPE) is reasonable for comparing reactivity.

Table R6. The calculated reaction energies ($E1$, $E2$, $E3$ unit: eV) of $R1$, $R2$, $R3$ on Pt(111), Pt₄@Gr, Pt₁@Gr catalysts with PBE+ZPE and PBE+D3+ZPE. $R1$, $R2$, and $R3$ is respectively represented the reaction steps of 2,4-DNT = 2,4-DNT*, H₂+2,4-DNT* = H₂*+2,4-DNT* and H₂*+2,4-DNT* = 2H*+2,4-DNT*.

	PBE+ZPE			PBE+D3+ZPE		
	$E1$	$E2$	$E3$	$E1$	$E2$	$E3$
Pt(111)	-0.23 eV	/	-1.13 eV	-1.41 eV	/	-1.30 eV
Pt ₄ @Gr	-0.27 eV	-0.16 eV	-0.87 eV	-1.39 eV	-0.10 eV	-0.74 eV
Pt@Gr	-0.53 eV	0.01 eV	0.18 eV	-1.71 eV	-0.28 eV	0.47 eV

REVIEWER COMMENTS

Reviewer #1 (Remarks to the Author):

The authors have satisfactory answer to most of my comments. Before possible submission I still have some concerns considering metal single atoms coordination and possible H-spillover.

Considering the preparation route of this material (high temperature heat-treatment under inert and then air exposure at room temperature, the material should contain significant amount of surface oxygen functional groups. And this has indeed been reported by the authors in 10.1038/s41467-019-12460-7 (fig S4) and <https://doi.org/10.1021/jacs.8b07476> (figure s2). The presence of these groups should be at least commented, but also considered when discussing single atom coordination (XAS does not make the distinction between carbon and oxygen).

The presence of these groups should also be considered for H-spillover, as different works have shown that these groups are the shuttles for H-spillover. Thus these groups should be considered in the DFT calculations (Tables R1 and R2). Moreover, simple experiments (maybe shorter than calculations) with WO₃ can show the presence or not of H-spillover.

Reviewer #2 (Remarks to the Author):

All my comments have been addressed in a professional and convincing manner.

Reviewer #3 (Remarks to the Author):

The submitted reviewers' rebuttal does not answer the reviewers' questions and comments. Besides, the answers to the Reviewers should be integrated into the manuscript wherever suitable.

Reviewer#1Q1 does not ask for the properties of the ND@G material but for the label ND@G.

In Reviewer#1Q3 and Q5, the authors made a significant number of extra calculations but did not address the reviewer's question. It has been shown that C-vacancies anchor H atoms (DOI: 10.1021/acs.jpcc.1c03996). It is evident that CH₃OH will not take another H to transfer it. However, CH₃O- could get the H from Pt₄ and bring it to Pt₁, which has not been considered. Why is the energy of H*(at C far from Pt₄ site) and H*(at C far from Pt SP site) so different (1.34 vs 86 eV)? Statistical analysis would reveal the H's probability of moving from Pt₄ to the C surface to Pt₁, which is ~0.3 eV more stable and thermodynamically driven.

Reviewer#1Q4. (1) Raman spec will also reveal vacancies in the particle core, meaning the resulting evaluation is unsuitable. (1) The Pt₁Cl₂ should be compared with the Pt₄Cl_x model, as the Pt₄ coordination also indicates monovacancy; the Eads difference in the presence of Cl is significant.

Reviewer#1Q7. The Eads value for 2,4-DAT is significantly larger than the intermediates on Pt(111), which is strange considering that 2,4-DNT adsorption is weaker than on Pt₄. There is no schematic representation or further details of these surface structures. Being a crucial part of the discussion, these structures should be somewhere in the paper, SI preferentially.

Answers regarding nanoparticles in #1Q7 and commercial nanoparticles in #2Q5 should be interpreted and critically discussed in the manuscript.

Reviewer#3Q2's answer says, "Another advantage is a small contact angle between the metal and support, forming a layered structure, which enhances the interaction between metal atoms and support, and ultimately increases the stability of the clusters." If the structures are layered, the DFT

model for Pt4 is wrong. Layered structures are different from FECC.

Reviewer#3Q9. Since when is eV the unit of charge?

Reviewer#3Q10. (1) Based on H₂ adsorption and dissociation energies and upon flushing the reactor 3 times with H₂, the catalyst will inevitably have H-adatoms on the surface. If the catalysts were saturated with 2,4-DNT, as said in the response, there would be no sites for H₂ to dissociate, so the reaction would not proceed. (2) What does it mean that the adsorption energies are over-corrected? How is it possible that long-range interaction increases the reaction energy (E₁) from -0.23 to -1.41 eV compared with these on E₂ and E₃? – This question remains unanswered and, in agreement with #3, it is critical for the evaluation of the models.

REVIEWER COMMENTS

Reviewer #1 (Remarks to the Author):

Comment. The authors have satisfactory answer to most of my comments. Before possible submission I still have some concerns considering metal single atoms coordination and possible H-spillover. Considering the preparation route of this material (high temperature heat-treatment under inert and then air exposure at room temperature, the material should contain significant amount of surface oxygen functional groups. And this has indeed been reported by the authors in 10.1038/s41467-019-12460-7 (fig S4) and <https://doi.org/10.1021/jacs.8b07476> (figure s2). The presence of these groups should be at least commented, but also considered when discussing single atom coordination (XAS does not make the distinction between carbon and oxygen). The presence of these groups should also be considered for H-spillover, as different works have shown that these groups are the shuttles for H-spillover. Thus, these groups should be considered in the DFT calculations (Tables R1 and R2). Moreover, simple experiments (maybe shorter than calculations) with WO_3 can show the presence or not of H-spillover.

Response: We thank the reviewer and appreciate the insightful comment and suggestion, which has helped us much in improving the quality of our manuscript. Although we have satisfactory answer to most of his/her comments, we are still sorry that some points are not well addressed in the 1st-round response. We would like to address this comment from the following aspects.

(1) As the reviewer mentioned, our previous paper has reported that the ND@G

inevitably contain some surface oxygen functional groups. XPS spectra of the ND@G confirmed that the oxygen functional groups including C-O (oxygen singly bonded to aliphatic carbon), C=O (oxygen doubly bonded to aromatic carbon), and a small number of phenolic (oxygen singly bonded to aromatic carbon). In our study, Pt atoms were loaded onto the ND@G surface and subsequently subjected them to reduction in H₂ at 200 °C. Following the reduction treatment, a significant decrease in oxygen content was observed (from about 3.9% down to approximately 2.5%), indicating a remarkable reduction in the content of the relevant oxygen functional groups. We agree with the reviewer that the oxygen functional group could potentially serve as the shuttles for H-spillover. However, due to the limited presence of oxygen functional groups in our research system, the feasibility of hydrogen transfer through these groups is diminished.

(2) According to the fitting results of EXAFS, due to the difficulty in distinguishing between O and C atoms in EXAFS, it is speculated that the structure may include oxygen coordinated structures, namely Pt₁O₂@Gr and Pt₁O₁C₁@Gr. We calculated the energies of possible structures of Pt₁O₂@Gr (Table R1) and Pt₁O₁C₁@Gr (Table R2). As a result, it was found that Pt₁O₂@Gr and Pt₁O₁C₁@Gr has a possible stable structure of Pt₁O₂@Gr-(2) and Pt₁O₁C₁@Gr-(6) respectively. Considering the reducing atmosphere in the reaction (25 °C, 1 MPa H₂), the oxygen in the oxygen-containing structure is likely to be reduced by H₂ to form H₂O. Therefore, we considered the thermodynamic possibilities of the following reactions.

The Gibbs free energy of H_2 , H_2O gas species ($T = 25\text{ }^\circ\text{C}$, $P(\text{H}_2) = 1\text{ MPa}$, $P(\text{H}_2\text{O}) = 0.1\text{ MPa}$) is calculated by VASPKIT code (Computer Physics Communications, 2021, 267, 108033.) and the Gibbs free energy of slabs is approximately equal to the DFT derived energy. Table R3 and Table R4 show the calculated Gibbs free energy of species and reaction Gibbs free energy, respectively. The continuous reduction of $\text{Pt}_1\text{O}_2@\text{Gr}$ (2), $\text{Pt}_1\text{O}_1@\text{Gr}$ -(6) and $\text{Pt}_1@\text{Gr}$ is exergonic by -1.58 eV , indicated the removal of surface oxygen and the formation of $\text{Pt}_1@\text{Gr}$ is thermodynamically favored. Therefore, the $\text{Pt}_1@\text{Gr}$ model is chosen to present the possible working catalyst for investigating both the adsorption and reaction process.

(3) In response to the reviewer's suggestion, we performed the WO_3 experiments (Figure R1). The inherent color of pure WO_3 is canary yellow, whereas the combination of catalysts and WO_3 exhibits a dark yellow. The color of WO_3 remained unchanged in the presence of H_2 . After exposing the mixture of $\text{Pt}_n/\text{ND}@G$ and WO_3 to the H_2 atmosphere, no significant hydrogen spillover phenomenon was observed.

In summary, in our research system, we propose that Pt_4 clusters play a major role in the hydrogenation process of 2,4-DNT. H_2 dissociates at the Pt_4 sites where the reactants are adsorbed, preferring to react directly with 2,4-DNT at the Pt_4 sites, enhancing the overall efficiency of the reaction.

Table R1. The possible structures of Pt₁O₂@Gr. The energy (DFT calculation) is given to compare the relative stabilities of these structures.

Entry	Before optimization	After optimization	Energy (eV)
1			-662.77
2			-666.35
3			-664.77
4			-665.06
5			-661.69
6			-660.56

Table R2. The possible structures of Pt₁O₁@Gr. The energy (DFT calculation) is given to compare the relative stabilities of these structures.

Entry	Before optimization	After optimization	Energy (eV)
1			-657.50
2			-656.82
3			-656.93
4			-657.36
5			-656.39
6			-660.02

Table R3. The Gibbs free energy of species in reaction (1) and (2).

Species	E_{DFT} (eV)	E_{ZPE} (eV)	G (eV)
H ₂ (g)	-6.76	0.28	-6.75
H ₂ O (g)	-14.22	0.60	-14.13
Pt ₁ O ₂ @Gr-(2)	-666.35	/	-666.35
Pt ₁ O ₁ @Gr-(6)	-660.02	/	-660.02
Pt ₁ @Gr	-653.16	/	-653.16

Table R4. The reaction Gibbs free energy of reaction (1) and (2).

	ΔE (eV)	ΔG (eV)
Reaction (1)	-0.81	-1.06
Reaction (2)	-0.27	-0.52

Figure R1. Photographs of samples made of WO₃ mixed with the catalysts before treatment (0 s) and after treatment with H₂ at 25 °C for 60 s.

Reviewer #2 (Remarks to the Author):

All my comments have been addressed in a professional and convincing manner.

Response: We sincerely appreciate the reviewer for the comments, which help us to improve the quality of our manuscript. We are glad that the responses are satisfactory to you. Thanks very much for your insightful comments again.

Reviewer #3 (Remarks to the Author):

The submitted reviewers' rebuttal does not answer the reviewers' questions and comments. Besides, the answers to the Reviewers should be integrated into the manuscript wherever suitable.

Response: We would like to appreciate the referee for reviewing our manuscript. We are really sorry that some points are not well addressed in the 1st-round response. We give more responses regarding the aspects your respected reviewer concerned. The relevant discussions were added in the revised manuscript.

Comments 1. Reviewer#1Q1 does not ask for the properties of the ND@G material but for the label ND@G.

Response: We thank the reviewer for the nice suggestion. The ND@G support is composed of a nanodiamond core and a defective graphene nanoshell. As shown in Figure R2, the core of each particle consists of a well-defined crystalline nanodiamond (ND) with (111) planes covered by an outer shell of defective amorphous graphene carbon (G). Hence, we named this support as nanodiamond@graphene (ND@G).

Figure R2. (a) HRTEM image of the as-prepared ND@G support with nanodiamond core and defective graphene shell; (b) Structure diagram of ND@G.

Comments 2. In Reviewer#1Q3 and Q5, the authors made a significant number of extra calculations but did not address the reviewer's question. It has been shown that C-vacancies anchor H atoms (DOI: 10.1021/acs.jpcc.1c03996). It is evident that CH₃OH will not take another H to transfer it. However, CH₃O⁻ could get the H from Pt₄ and bring it to Pt₁, which has not been considered. Why is the energy of H*(at C far from Pt₄ site) and H*(at C far from Pt SP site) so different (1.34 vs 0.86 eV)? Statistical analysis would reveal the H's probability of moving from Pt₄ to the C surface to Pt₁, which is ~0.3 eV more stable and thermodynamically driven.

Response: We really appreciate the reviewer's insightful comments and valuable suggestions, which really help us a lot in improving the manuscript. We are sorry that we didn't write clearly enough in the previous version. We have provided the definitions of the H* sites and the corresponding structure of H-transfer by carbon defect sites (Figure R3) and H-transfer by CH₃OH (Figure R4). We have added a pathway for the formation of methoxy groups from methanol on Pt₄@Gr and compared it with other H-spillover routes. And the results shows (Figure R4) that the formation of methoxy on Pt₄@Gr is not favored kinetically (0.94 eV) in the current temperature (25 °C). The migration of H* from Pt₄@Gr to Pt₁@Gr is overall exothermic by 0.30 eV, indicating stronger adsorption of H* on Pt₁@Gr. However, by comparing H-spillover routes, including CH₃OH as the H carrier, surface carbon defect sites, and the *in situ* direct hydrogenation of 2,4-DNT on Pt₄@Gr, it is found that *in situ* hydrogenation is continuous and strongly exothermic, which may cause a rapid consumption of H and will be the most advantageous route.

Figure R3. H-spillover by defected carbon sites.

Figure R4. H-spillover by CH₃OH carrier.

Comments 3. Reviwer#1Q4. (1) Raman spec will also reveal vacancies in the particle core, meaning the resulting evaluation is unsuitable. (2) The Pt₁Cl₂ should be compared with the Pt₄Cl_x model, as the Pt₄ coordination also indicates monovacancy; the Eads

difference in the presence of Cl is significant.

Response: We thank you very much for your constructive comments, which help us to improve the quality of our manuscript. We would like to address this comment from the following aspects.

(1) As the reviewer mentioned, Raman spectroscopy can reveal characteristic peaks in the ND. Indeed, previous literature has reported that ND and ND@G exhibit distinct characteristic peaks in Raman spectra (Figure 2. Chem. Eur. J., 2014, 20, 6324-6331. <https://doi.org/10.1002/chem.201400018>). The spectrum of ND displayed a broadened sharp diamond peak at 1324 cm^{-1} and a broad peak with a maximum at about 1635 cm^{-1} . For ND@G samples (referred to as ND-1100 in this literature), annealing at 1100°C results in rapid weakening of the diamond peak which overlaps with a new broad peak at approximately 1400 cm^{-1} assigned to disorder-induced D band (originating from defects present in sp^2 -hybridized carbon); furthermore, there was also a shift of a broad peak from 1635 cm^{-1} to approximately 1580 cm^{-1} , which can be interpreted as well-ordered nanocrystalline graphite (G band). In the case of the ND samples, the D- and G-peaks of carbon were not observed in Raman spectra. The intensity ratio of $I_{\text{D}}/I_{\text{G}}$, which is normally used to qualitatively estimate the defect degree of sp^2 -bonded carbon materials, was found to be 0 for the ND samples and approximately 0.76 for the ND@G samples according to this literature report. The results show that the signals of characteristic peaks of ND samples in Raman spectra do not exert any influence on the resulting evaluation; thus, the $I_{\text{D}}/I_{\text{G}}$ ratio serves as a reliable indicator of carbon defects on the surface of ND@G.

(2) Indeed, in the EXAFS results, there was no direct Pt–Cl coordination in Pt_n/ND@G. Based on the reviewer's suggestions, we have added calculations for the adsorption of 2,4-DNT, 2A4NT, 2N4AT, and 2,4-DAT on the Pt₄Cl₂@Gr and provided the corresponding structure to compare with Pt₁Cl₂@Gr (Table R5). And the results showed that the adsorption of all intermediates was difficult to occur due to Cl steric hindrance (0.11, 0.18, 0.20, and 0.20 eV). Therefore, we believe that Pt₄Cl₂@Gr is reactive inert and that the Pt₄ coordinated with C atom is the working active site.

Table R5. The optimized structure and adsorption energy (E_{ads}) of 2,4-DNT, 2N4AT, 2A4NT, and 2,4-DAT on PtCl₂@Gr and Pt₄Cl₂@Gr.

Name	Structure	E_{ads} (eV)	Name	Structure	E_{ads} (eV)
PtCl ₂ @Gr		/	Pt ₄ Cl ₂ @Gr		/
2,4-DNT		-0.09	2,4-DNT		0.11
2N4AT		-0.09	2N4AT		0.18
2A4NT		-0.13	2A4NT		0.20
2,4-DAT		-0.22	2,4-DAT		0.20

Comments 4. Reviewer#1Q7. The Eads value for 2,4-DAT is significantly larger than the intermediates on Pt(111), which is strange considering that 2,4-DNT adsorption is weaker than on Pt4. There is no schematic representation or further details of these surface structures. Being a crucial part of the discussion, these structures should be somewhere in the paper, SI preferentially.

Response: We thank the reviewer for the constructive comment, which help us to improve the quality of our paper a lot. We are very sorry that we didn't give the relevant DFT model structure in the former manuscript. According to the reviewer's comments, we added schematic of these structures, the optimized structure and energy, as well as zero-point energy (ZPE) corrected values (Figure R5, Table R6-8). It is found that there is slight difference of the adsorption energy of 2,4-DNT on Pt₄@Gr and Pt(111) (by 2-NO₂-group: -0.26 vs. -0.18 eV; by 4-NO₂-group: -0.24 vs. -0.19 eV; by benzene-group: -0.23 vs. -0.27 eV). In fact, we think that these configurations exist in thermodynamic equilibrium. For 2A4NT and 2,4-DAT, both of them are in the configuration of benzene-groups on Pt₄@Gr and on Pt (111). On Pt₄@Gr, the adsorption energy of 2A4NT increased by 0.18 eV compared with 2,4-DNT, and the adsorption energy of 2,4-DAT increased by 0.29 eV compared with 2A4NT. The increasing trend but different amplification also was found on Pt (111), with corresponding values of 0.57 and 0.77 eV. We think that this is because the interaction between different coordinated environment [Pt₄@Gr vs. Pt (111)] and NO₂-/NH₂- modulated π bond of benzene. To make the DFT part more clearly and readable, we added these structures and discussion in the revised manuscript. “Further calculation was carried out to study the adsorption

behavior of intermediates and products on Pt₄@Gr in comparison with Pt(111). The involved structure and adsorption energies were also supplied in SI (Supplementary Table 7-9). For 2A4NT and 2,4-DAT, both of them are in the configuration of benzene-groups on Pt₄@Gr and on Pt (111). Compared Pt(111) with Pt₄@Gr, it is found that although H₂ dissociation is easier on Pt(111), the adsorption of 2,4-DAT is too strong (-1.57 vs. -0.74 eV, Supplementary Table 7), which makes the intermediates and products accumulation on active sites and causes a decrease in activity.” (Please see Line 338-346)

Figure R5. Three possible adsorption models (at 2-NO₂- configuration, at 4-NO₂- configuration and benzene- configuration) and the corresponding adsorption energy of 2,4-DNT over the Pt₄@Gr and Pt(111).

Table R6. The adsorption energies E_{ads} (eV) of intermediates on Pt(111), Pt₄@Gr and Pt₁@Gr, respectively.

	Pt(111)	Pt ₄ @Gr	Pt ₁ @Gr
2,4-DNT	-0.23	-0.27	-0.53
2N4AT	-0.73	-0.44	-0.63
2A4NT	-0.80	-0.45	-0.67
2,4-DAT	-1.57	-0.74	-0.84

Table R7. The optimized structure, DFT energy (E_{DFT}) and corrected zero-point energy (E_{ZPE}) of 2,4-DNT, 2N4AT, 2A4NT, and 2,4-DAT on Pt (111)

	Structure	E_{DFT} (eV)	E_{ZPE} (eV)
2,4-DNT		-956.84	3.48
2N4AT		-954.29	3.98
2A4NT		-954.46	3.92
2,4-DAT		-951.89	4.36

Table R8. The optimized structure, DFT energy (E_{DFT}) and corrected zero-point energy (E_{ZPE}) of 2,4-DNT, 2N4AT, 2A4NT, and 2,4-DAT on Pt₄@Gr.

	Structure	E_{DFT} (eV)	E_{ZPE} (eV)
2,4-DNT		-762.75	3.49
2N4AT		-759.80	3.91
2A4NT		-759.95	3.90
2,4-DAT		-756.87	4.32

Comments 5. Answers regarding nanoparticles in #1Q7 and commercial nanoparticles in #2Q5 should be interpreted and critically discussed in the manuscript.

Response: We thanks very much for your constructive comments, which help us to improve the quality of our manuscript. Following the reviewer’s advice, we have added discussions in the revised manuscript. “As shown in Supplementary Fig. 3, clear Pt nanoparticles were visible in all catalysts through STEM images.” (Please see Line 228-229) “The above results further indicate that the ND@G support is more efficient in dispersing and stabilizing metal atoms compared to oxides or carbon materials.” (Please see Line 234-236) “Furthermore, the commercial 5 wt% Pt/C was used as a reference

sample, which displayed well dispersed Pt nanoparticles (Supplementary Fig. 4). As shown in Figure 3c, the commercial Pt/C catalyst displayed lower activity compared to Pt_n/ND@G.” (Please see Line 236-239)

Comments 6. Reviewer#3Q2’s answer says, “Another advantage is a small contact angle between the metal and support, forming a layered structure, which enhances the interaction between metal atoms and support, and ultimately increases the stability of the clusters.” If the structures are layered, the DFT model for Pt₄ is wrong. Layered structures are different from FECC.

Response: We thank the reviewer for the nice suggestion. We are sorry that we didn’t write clearly enough in our former manuscript. The claim of layered metal clusters means that there are no metal atoms covering each other and all metal atoms in the clusters can get exposed to reactants during reaction. To avoid the potential confusion and clarify our “FECC” conception more clearly, we have revised the relevant discussion in the manuscript. “Supported FECCs have two distinct advantages: their ultrasmall size (typically below 1 nm) eliminates undesired bulk atoms and reduces the average coordination number of metal atoms, while a small contact angle between the metal and support enhances interaction and ultimately increases cluster stability.” (Please see Line 100-103)

Comments 7. Reviewer#3Q9. Since when is eV the unit of charge?

Response: We thank the reviewer for pointing out this for us. We have carefully

checked the text and corrected this mistake in the revision manuscript.

Comments 8. Reviewer#3Q10. (1) Based on H₂ adsorption and dissociation energies and upon flushing the reactor 3 times with H₂, the catalyst will inevitably have H-adatoms on the surface. If the catalysts were saturated with 2,4-DNT, as said in the response, there would be no sites for H₂ to dissociate, so the reaction would not proceed. (2) What does it mean that the adsorption energies are over-corrected? How is it possible that long-range interaction increases the reaction energy (E1) from -0.23 to -1.41 eV compared with these on E2 and E3? – This question remains unanswered and, in agreement with #3, it is critical for the evaluation of the models.

Response: We sincerely appreciate the reviewer for the insightful comment and constructive suggestion. We are sorry that we didn't write clearly enough in the previous version. We would like to address this comment from the following aspects.

(1) As mentioned by the reviewer, the saturation of the adsorbed reactants at the active site was an adsorption-desorption equilibrium process. It is possible that a few H-adatoms may unavoidably be present on the surface of catalysts during the flushing with H₂ and subsequent introduction of H₂. However, due to the dissimilar solubilities of 2,4-DNT and H₂, it is noteworthy that the concentration of 2,4-DNT in methanol is approximately three times higher than that of H₂ in this liquid-phase hydrogenation reaction. Therefore, a majority of active sites are still saturated with adsorbed 2,4-DNT. Pt single atom catalysts suffer from a lack of sufficient active sites to dissociate H₂, resulting in limited catalytic activity. In contrast, Pt cluster catalysts displayed excellent

catalytic activity due to the enough of active sites.

(2) $E1$ and $E2$ involve the energy difference before and after the adsorption of free molecular, while $E3$ is the energy difference of surface reactions (Figure R6, Table R9). Because only surface species were subjected to van der Waals correction, the changes in $E3$ were not significant. For $E1$ and $E2$, $E1$ involves the adsorbed 2,4-DNT, while $E2$ involves the adsorbed H_2 . The van der Waals correction value will be greater for 2,4-DNT than for H_2 , as a result, $E1$ undergoes significant changes. We performed single point calculations on the PBE optimized structure using methods of D2 and D3 correction, and the obtained energy is shown in Table R9-10. Compared with PBE-derived adsorption energy, the adsorption energy by D2 and D3 methods is more negative, but they have the same trend. We correct our statement about over-correction and believe that PBE+ZPE, PBE+D2+ZPE, and PBE+D3+ZPE have reached the same conclusion that the reactivity of $Pt_1@Gr$ is limited by the adsorption of H_2 .

Figure R6. The calculated reaction energies (E_1 , E_2 , E_3) of R_1 , R_2 , R_3 on Pt(111), Pt₄@Gr, Pt₁@Gr catalysts with PBE+ZPE. R_1 , R_2 , and R_3 is respectively represented the reaction steps of $2,4\text{-DNT} = 2,4\text{-DNT}^*$, $\text{H}_2+2,4\text{-DNT}^* = \text{H}_2^*+2,4\text{-DNT}^*$ and $\text{H}_2^*+2,4\text{-DNT}^* = 2\text{H}^*+2,4\text{-DNT}^*$.

Table R9. The calculated reaction energies (E_1/eV , E_2/eV , E_3/eV) of R_1 , R_2 , R_3 on Pt(111), Pt₄@Gr, Pt₁@Gr catalysts with PBE+ZPE, PBE+D2+ZPE and PBE+D3+ZPE. R_1 , R_2 , and R_3 is respectively represented the reaction steps of $2,4\text{-DNT} = 2,4\text{-DNT}^*$, $\text{H}_2+2,4\text{-DNT}^* = \text{H}_2^*+2,4\text{-DNT}^*$ and $\text{H}_2^*+2,4\text{-DNT}^* = 2\text{H}^*+2,4\text{-DNT}^*$.

	PBE+ZPE			PBE+D2+ZPE			PBE+D3+ZPE		
	E_1	E_2	E_3	E_1	E_2	E_3	E_1	E_2	E_3
Pt(111)	-0.23	/	-1.13	-1.82	/	-1.19	-1.41	/	-1.30
Pt ₄ @Gr	-0.27	-0.16	-0.87	-1.31	-0.19	-0.75	-1.39	-0.10	-0.74
Pt ₁ @Gr	-0.53	0.01	0.18	-1.52	-0.30	0.50	-1.71	-0.28	0.47

Table R10. PBE-derived energy (E_{PBE}/eV), single point energy corrected by D2 correction (E_{D2}/eV) and D3 correction (E_{D3}/eV), and corrected zero-point energy (E_{ZPE}) of reaction intermediates.

	Structure	E_{PBE}	E_{D2}	E_{D3}	E_{ZPE}
Pt(111)					
2,4-DNT*		-956.84	-1128.72	-1038.79	3.48
$\text{H}_2^* + 2,4\text{-DNT}^*$	/	/	/	/	/
$2\text{H}^* + 2,4\text{-DNT}^*$		-964.81	-1136.73	-1046.90	3.83
Pt(111)		-833.35	-1003.61	-913.91	/
Pt ₄ @Gr					
2,4-DNT*		-762.75	-768.96	-770.99	3.49
$\text{H}_2^* + 2,4\text{-DNT}^*$		-769.72	-775.88	-777.82	3.82
$2\text{H}^* + 2,4\text{-DNT}^*$		-770.62	-776.67	-778.59	3.85

Pt ₄ @Gr		-666.17	-644.30	-646.24	/

Pt ₁ @Gr					

2,4-DNT*		-777.04	-782.24	-784.70	3.46
H ₂ *+2,4-DNT*		-783.86	-789.36	-791.80	3.80
2H*+2,4-DNT*		-783.81	-788.99	-791.45	3.93
Pt ₁ @Gr		-653.16	-657.47	-659.74	/

Reaction species					

H ₂		-6.76	/	/	0.28
2,4-DNT		-123.32	/	/	3.53

REVIEWERS' COMMENTS

Reviewer #1 (Remarks to the Author):

The authors of this work have made a significant effort to satisfactorily answer the questions asked. This article may be published.

REVIEWERS' COMMENTS

Reviewer #1 (Remarks to the Author):

The authors of this work have made a significant effort to satisfactorily answer the questions asked. This article may be published.

Response: Many thanks to the Reviewer #1 for the previous valuable comments and publication recommendation on this work, which help us to improve the quality of our manuscript. We are glad that the responses are satisfactory to you. Thanks very much for your insightful comments again.